# Shifts in honeybee foraging reveal historical changes in floral resources

Laura Jones 1,2, Georgina L. Brennan[3], Abigail Lowe[1,2], Simon Creer 2, Col R. Ford 1 & Natasha de Vere 1,4✉

Decreasing floral resources as a result of habitat loss is one of the key factors in the decline of pollinating insects worldwide. Understanding which plants pollinators use is vital to inform the provision of appropriate floral resources to help prevent pollinator loss. Using a globally important pollinator, the honeybee, we show how changes in agricultural intensification, crop use and the spread of invasive species, have altered the nectar and pollen sources available in the UK. Using DNA metabarcoding, we analysed 441 honey samples from 2017 and compared these to a nationwide survey of honey samples from 1952. We reveal that shifts in major plants foraged by honeybees are driven by changes in the availability of these plants within the landscape. Improved grasslands are the most widespread habitat type in the UK, and management changes within this habitat have the greatest potential to increase floral resource availability.

[1] National Botanic Garden of Wales, Llanarthne SA32 8HG, UK. [2] MEFGL, School of Natural Sciences, Bangor University, Bangor LL57 2UW, UK. [3] Centre for Environmental and Climate Research / Aquatic Ecology, Lund University, 223 62 Lund, Sweden. [4] IBERS, Aberystwyth University, Aberystwyth SY23 3FL, UK. ✉email: natasha.devere@gardenofwales.org.uk

Widespread declines in insect pollinators and the associated impacts on crops and biodiversity is a global problem[1-3]. One of the major factors implicated in pollinator declines is the reduction of floral resources due to agricultural intensification and habitat loss[4-7]. Across Europe, the dominant land use is agriculture, with over half of the European landscape being managed agriculturally[8]. Agricultural land is therefore a focus of conservation efforts to prevent the loss of associated biodiversity[8,9]. The input of inorganic nitrogen, re-seeding of grassland leys, high levels of grazing, and herbicide application can all cause species-rich, semi-natural grassland to become improved grassland, with a corresponding reduction in the diversity and availability of flowers used by pollinating insects for nectar and pollen[10]. In England and Wales, the proportion of lowland semi-natural grassland has been estimated to be 3% of what was present prior to 1939[11].

It is possible to track changes in the availability of nectar resources over time by combining vegetation surveys and direct nectar measurements[12], but it is difficult to relate this to changes in pollinator foraging. Honeybees are an ideal model to assess landscape changes in forage availability and usage as they have a widespread distribution and long foraging range[13]. Managed honeybees can be geolocated exactly and so by characterising the pollen found within honey, we can determine the floral resources used for nectar and pollen in the area surrounding the hive[14].

Here, honey provided by beekeepers as part of a nationwide UK campaign in 2017 has been characterised and compared to honey sampled in 1952, enabling us to investigate whether landscape-scale changes in the floral resource are leading to changes in honeybee foraging. DNA metabarcoding, using two complementary DNA barcode markers (*rbcL* and ITS2), was used to identify the plant taxa within contemporary honey samples, extracted from hives between April and October, across the latitudinal and longitudinal range of the UK in 2017. We compared the plant composition of the 2017 honey with the last UK wide survey of honey samples from 1952, characterised using melissopalynology[15,16]. DNA metabarcoding leverages a higher taxonomic resolution of the plant taxa present in the honey when compared to microscopic identification and so a conservative approach was taken to compare the data.

## Results
In 2017, we analysed 441 honey samples, with most samples provided from England and Wales in July (147 samples) and August (155 samples) (Fig. 1). The habitat type surrounding the hives reflected the composition of habitats of the UK with a positive correlation between the proportion of habitats in the UK and the proportion in a 2-km radius around the hives (Fig. 1, $r^2 = 0.8$, $P = 0.0002$). Improved grassland, arable and horticulture, broadleaved woodland and suburban were the top habitats within the locality of the hives (Fig. 1).

**UK honeybee foraging**. A total of 157 plant taxa were identified from the 441 honey samples, using the *rbcL* and ITS2 barcode regions combined (Supplementary Data). The total frequency of occurrence for each plant taxon was calculated as the presence of the taxon across all 2017 honey samples (Fig. 2). Of the 157 identified taxa, only 44 occurred in over 5% of the honey samples and only four taxa were identified in over 50% of samples (Fig. 2). For the abundance of each plant taxon within a sample, the proportion of sequences returned was placed into classes. Plant taxa represented by over 45% of sequences were designated predominant for that sample; between 15 and 45% were secondary; between 1 and 15% were important minor taxa and <1%

of reads were classed as minor taxa (Fig. 2). Major taxa were defined as taxa returned at a predominant or secondary level.

Brambles (*Rubus* spp.) were both the most frequently found and abundant species within the honey samples, followed by white clover (*Trifolium repens*) and *Brassica* species. The *Brassica* species include the crop, oilseed rape (*Brassica napus*), along with other wild and cultivated *Brassica* species. The high frequency of these taxa across the honey reflects their long flowering period, with these plant groups appearing at high levels from May to September (Fig. 3). The next most frequently found and abundant plants were spring-flowering shrubs and trees, including hawthorn (*Cratageus monogyna*), apple (*Malus* spp.), *Cotoneaster* spp., sycamore and maples (*Acer* spp.), and cherries and plums (*Prunus* spp.). Towards the end of the season (peaking in September), heather (*Calluna vulgaris*) and the non-native invasive species, Himalayan balsam (*Impatiens glandulifera*) were found abundantly within honey samples (Fig. 3). Pollen identified in honey collected in 2017 reflects the seasonal changes in the plants available to the honeybees, with calendar month (April–October), being a good predictor of plant taxa composition (Fig. 3; $LR_{428, 1} = 454.8$, $P = 0.001$).

There were no overall regional differences between England, Scotland and Wales, in the most frequently found taxa in 2017 (Supplementary Fig. 1; Latitude $LR_{427, 1} = 272.2$, $P = 0.086$; Longitude $LR_{426, 1} = 352.3$, $P = 0.092$). While latitude and longitude were not significant predictors when assessing the overall honey composition, at the individual taxa level there was some evidence of spatial autocorrelation in 22 of the 157 taxa identified (using Moran's I; Supplementary Data). However, after Bonferroni's correction for multiple testing none of the 22 taxa remained significant. The relationship between the plant composition of the honey and the dominant surrounding habitat class was significant however habitat class explained only 3% of the total variation (Supplementary Fig. 2; $r^2 = 0.037$, $P = 0.001$).

There were significant relationships between the distribution of insect attractive crops, field beans (*Vicia faba*) and oilseed rape (*Brassica napus*), and their presence within honey samples (Fig. 4). *Vicia* species were more likely to be detected in honey within a 2 km radius of field beans ($x^2 = 52.83$, d.f. = 4, $P < 0.001$) and *Brassica* species within 2 km of oilseed rape ($x^2 = 50.71$, d.f. = 4, $P < 0.001$).

**Comparision with 1952**. In 1952, 855 honey samples, from throughout the UK, were analysed using melissopalynology[15,16]. A total of 66 plant taxa were identified, 47 of which matched with the plants found in 2017 (Supplementary Fig. 3, Supplementary Data and Discussion).

Overall, there was a positive correlation between the frequency of occurrence for the 47 plant taxa between the two collection dates of 1952 and 2017 (Supplementary Fig. 4; Kendall's τ correlation coefficient τ = 0.389, $P < 0.001$). There were however, significant differences between 1952 and 2017 in the frequency of the major taxa, classed as predominant and secondary (Fig. 5; $LR_{125, 1} = 93.16$, $P = 0.001$), while no significant difference was found between sampling locations (Supplementary Fig. 5; $LR_{37, 88} = 508.0$, $P = 0.944$). Of the nine plant taxa returned as major taxa for honeybees in both 1952 and 2017, and present in over 1% of samples, seven of these plants show significant differences in use. This corresponds to differences in their frequency within the landscape as measured by the Countryside Survey (Fig. 5). The top forage found in 1952, white clover (*Trifolium repens*), was reported as a major plant in 74% of honey samples, decreasing to 31% in 2017 ($x^2 = 229.51$, d.f = 1, $P < 0.001$). Red clover (*Trifolium pratense*) also decreased in use from 5% of honey samples to 1% ($x^2 = 11.18$, d.f = 1, $P = 0.027$). Based on the Countryside Survey,

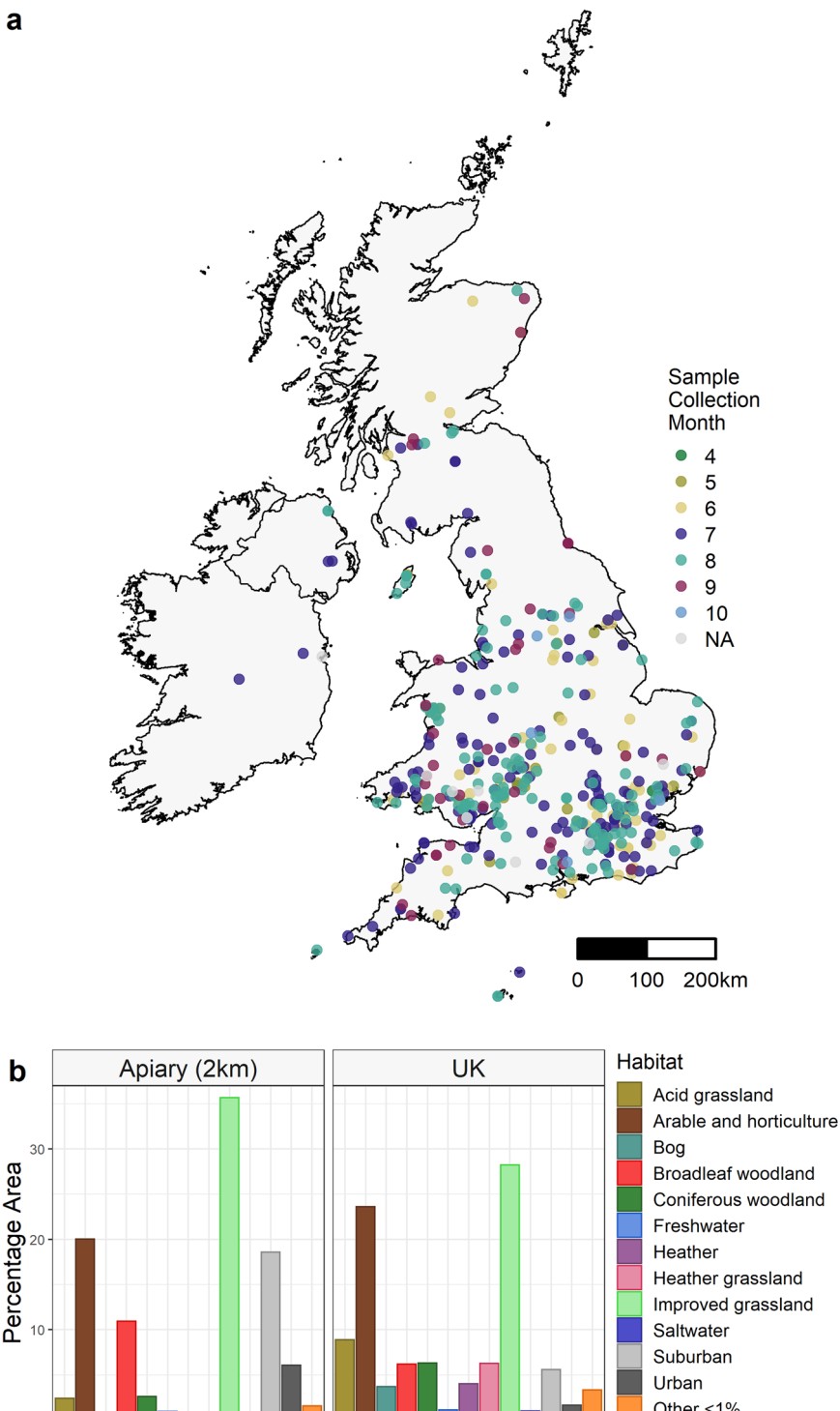

**Fig. 1 Distribution of honey samples ($n = 441$) collected in 2017 and analysed with DNA metabarcoding, along with the habitat types hives were found within. a** The month when honey was extracted from the hive, April (4) to October (10), is indicated by colours. **b** The percentage area of different habitats is presented for the UK as a whole and for within a 2-km radius of honey samples, characterised using the 2015 CEH Land Cover map (NERC CEH).

*Trifolium repens* decreased in the landscape by 13% and *T. pratense* by 27% between 1978 and 2007.

Contrasting the decline in the *Trifolium* species, brambles (*Rubus* spp.) have seen an increase in forage use compared to 1952 and are now the most foraged genus for honeybees in the UK. In 1952, *Rubus* was the major taxa in only 5% of honey samples, compared to 31% in 2017 ($x^2 = 367.07$, d.f $= 1$, $P < 0.001$), supported by the Countryside Survey which recorded an

increase in the most widely distributed and common species *Rubus fructicosus* by 21% between 1978 and 2007.

*Brassica* species were the major taxa source in only 1% of honey samples in 1952 compared with 21% in 2017 ($x^2 = 131.46$, d.f $= 1$, $P < 0.001$), which includes the insect attractive crop species oilseed rape (*Brassica napus*). No significant difference was found between the honey surveys for *Vicia* species, despite an increase in production of field beans (*Vicia faba*) since 1945

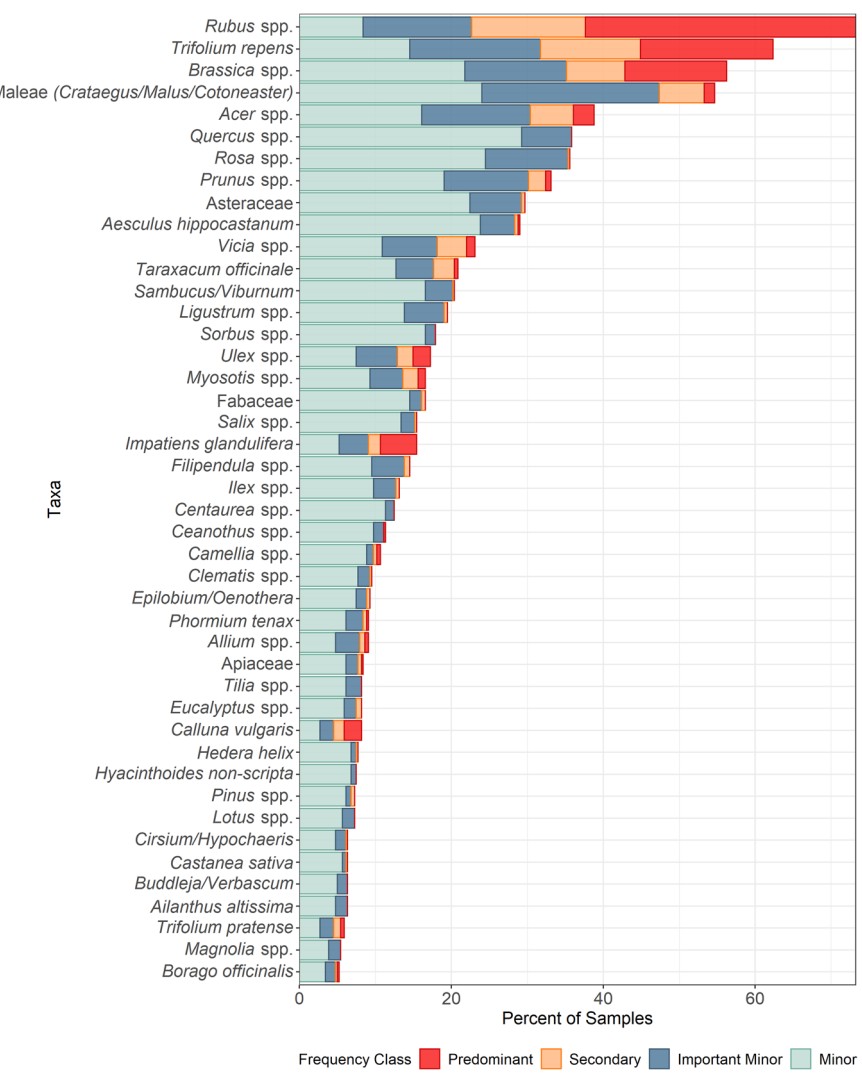

**Fig. 2 Plant taxa found in over 5% of honey samples analysed using DNA metabarcoding (*n* = 441).** The abundance of a plant taxon within a honey sample is indicated by the colour, with four abundance classes: Predominant: >45% of sequences returned in a sample, secondary: 15–45%, important minor: 1–15% and minor <1%.

$(x^2 = 7.15,\ \mathrm{d.f} = 1,\ P = 0.255)^{17–19}$. In contrast, the Countryside Survey shows a 26% decrease in *Vicia* species reflecting reductions in the availability of wild vetches.

Himalayan balsam (*Impatiens glandulifera*) is a non-native invasive species first introduced into the UK in 1839 which, after an initial lag phase, started to increase rapidly in distribution from 1940 to 1960[20,21]. *Impatiens glandulifera* increased as a major taxon from 1% of honey samples in 1952 to 6% in 2017 $(x^2 = 22.17,\ \mathrm{d.f} = 1,\ P < 0.001)$.

All analyses were additionally run on rarefied sequencing data (Supplementary Results). Using rarefied data did not change the conclusions from the statistical analyses completed here.

## Discussion

Improved grassland under agricultural management is the most widespread habitat of the UK (Fig. 1) and has been estimated to provide the greatest contribution to nationwide nectar resource, with *Trifolium repens* as the dominant source of nectar[12]. However, the presence of flowering *Trifolium* species has reduced substantially within managed grasslands, due to decreasing use of clover leys in crop rotation and the increased application of inorganic nitrogen fertilizers and herbicides[5,22]. Furthermore, the

clover that is present in modern grasslands may not be contributing to landscape estimates of nectar, as management that includes multiple cuts or intensive grazing can prevent flowering. Nevertheless, *T. repens* still represents the second most foraged plant in the 2017 survey suggesting that, despite declines in land cover, honeybees are still actively seeking out white clover.

White clover (*Trifolium repens*) and bramble (*Rubus fruticosus*) have similar flowering periods and the increased predominance of *Rubus* spp. within the honey may reflect the reduced availability of *Trifolium repens* during the same flowering period. Records show that *R. fruticosus* increased in local frequency between 1978 and 2007, however its distribution across the UK was not found to have changed between 1962 and 2001[23]. *R. fruticosus* and *T. repens* both offer pollen and nectar, however, the protein content and proportion of essential amino acids is lower in *R. fruticosus* compared to *T. repens*[24] meaning honeybees may not be gaining the same nutritional benefits if substituting *T. repens* with *R. fruticosus*.

A shift in landscape forage availability occurred with the increase of oilseed rape as a UK crop since the 1970s (4884 ha were grown in 1969 compared to 279,030 in 1988[5,25,26]), this is reflected in the increase in *Brassica* species within the 2017 honey samples. Field beans (*Vicia faba*) are another insect attractive

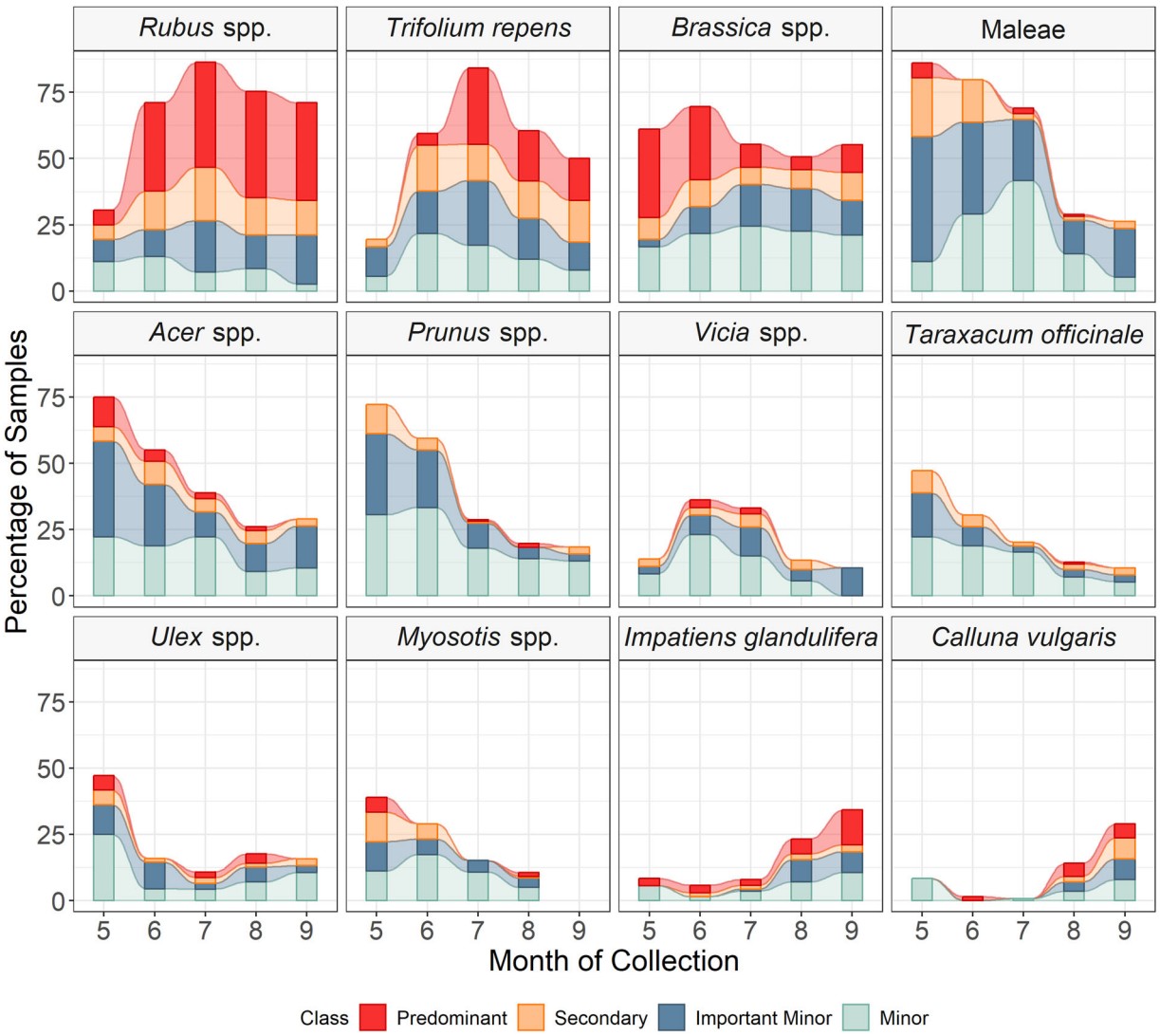

**Fig. 3 The most abundantly found plant taxa in the honey samples analysed using DNA metabarcoding are summarised as a proportion of samples through the season.** Plant taxa illustrated are the most frequently found plant taxa in 2017 at a predominant and secondary level. Samples collected in April ($n = 3$) and October ($n = 7$) were excluded. Sample sizes: May ($n = 39$), June ($n = 71$), July ($n = 147$), August ($n = 155$), September ($n = 43$). Predominant: >45% of sequences returned in a sample, secondary: 15–45%, important minor: 1–15% and minor <1%. The overall height of the bar indicates the total proportion of samples with that taxon.

species that have increased in production. However, the overall frequency of *Vicia* species within the honey was found to have decreased from 32 to 23%, reflecting the decline in wild vetches, which are a common component of species-rich grasslands[27]. As a major component of the honey, *Vicia* species have increased in frequency from 2 to 5%. with one possibility being that where there has been an increase in the availability of the field bean crop, honeybees will use them as a major honey source.

Himalayan balsam (*Impatiens glandulifera*) increased significantly as a major plant within the honey, representing a non-native invasive species that has increased in availability in the landscape. *I. glandulifera* is extremely attractive to pollinators with a higher nectar production when compared to other plant species associated with the same habitat[28]. Although attractive to pollinators, *I. glandulifera* can have a negative impact on native plant diversity by outcompeting other species for both space and pollinators, leading to a reduction in seed-set in co-occurring species[28].

While the dominant habitat class surrounding the hives showed a significant relationship with the plant composition of the honey, it explained a limited amount of the total variation.

The plants found here to be the most frequently used by honeybees are widely distributed in the UK. Honeybees may be selecting the same frequently found plants across different habitat classes, with the time of year being a better predictor for plant choice. Strong seasonal variation unrelated to the surrounding landscape diversity has been seen in the pollen collected by honeybees[29]. In addition, the foraging distances for pollen-collecting bees have been shown to vary both with the complexity of surrounding landscapes and season[30], suggesting honeybees may be increasing their foraging range for certain forage plants. While the overall plant composition of the honey was found to unrelated to the location of the hive, further work could investigate the potential geographic patterns present in the spatially restricted plant species found at lower levels within the honey.

Agricultural intensification, changes in crop species and the spread of non-native invasive plants all contribute to changes in the available forage for honeybees and have wider implications for pollinator habitat management, since the key taxa identified represent the plant species which provide the greatest abundance of nectar nationally within the UK[8].

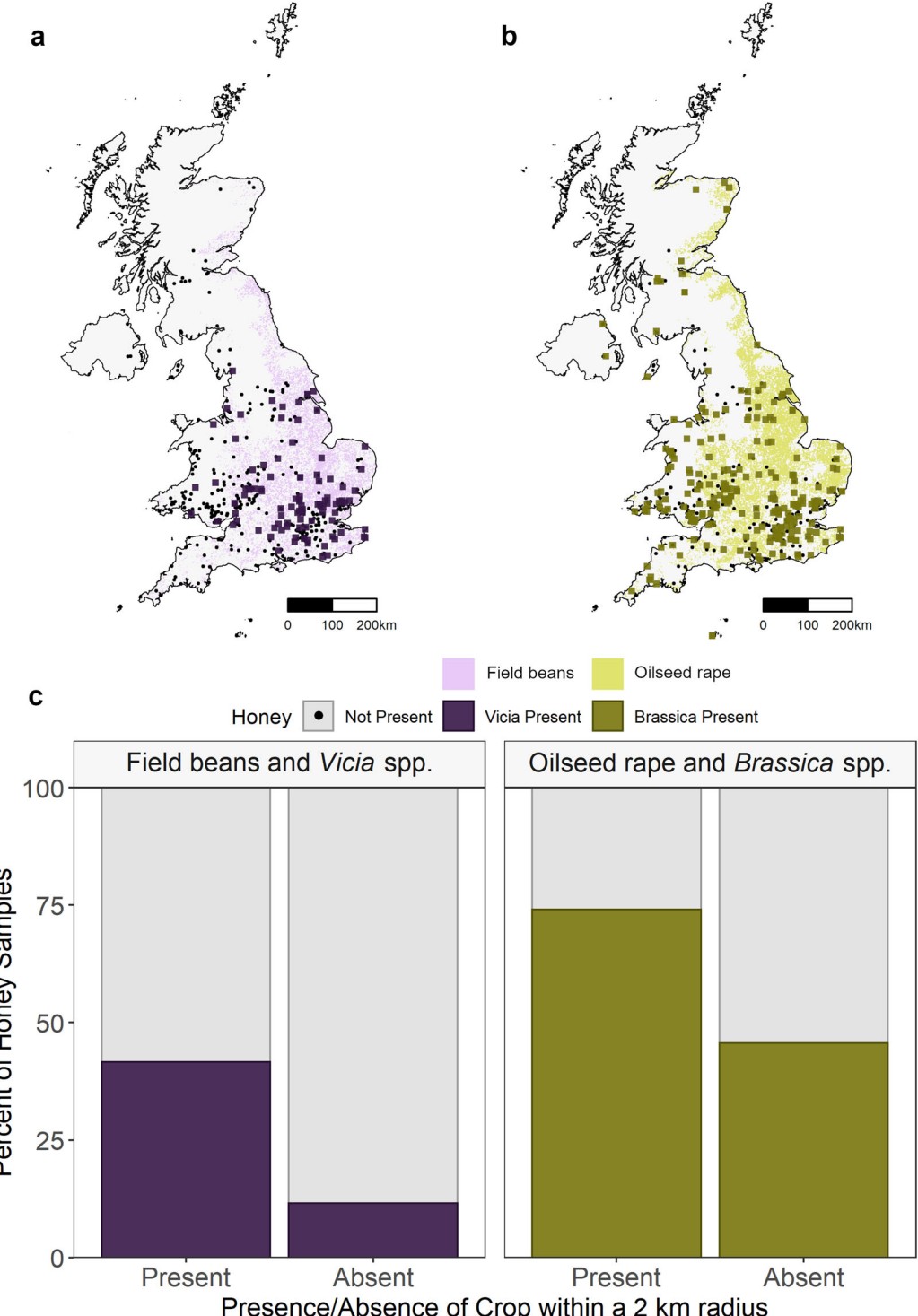

**Fig. 4 Distribution of field beans (*Vicia faba*) and oilseed rape (*Brassica napus*) in the UK and their presence within honey samples (*n* = 424) detected using DNA metabarcoding.** The distribution of field beans (**a**) and oil seed rape (**b**) in 2017 from CEH Land Cover plus: Crops map is shown under honey samples with *Vicia* spp. (**a**) and *Brassica* spp. (**b**) present, as indicated by coloured squares. Black circles: not detected. **c** *Vicia* spp. were more likely to be detected in honey within a 2 km radius of field beans ($x^2 = 52.83$, d.f. = 4, $p < 0.0001$) and *Brassica* spp. within 2 km of oil seed rape ($x^2 = 50.71$, d.f. = 4, $p < 0.0001$) (Supplementary Table 1).

On a landscape scale, the management recommendation that has the greatest potential to increase the quantity of nectar on a UK-wide basis is to increase the presence and diversity of nectar-rich species within improved grasslands, including flowering clover (*Trifolium repens* and *T. pratense*). Improved grasslands represent the most extensive habitat type of the UK and changes

to increase plant diversity and flower availability within this habitat will have the greatest impact on nectar and pollen provision.

Honeybees as a model provide an overview of the availability of these widespread foraging resources on an otherwise unachievable scale. However, this information should be set within the

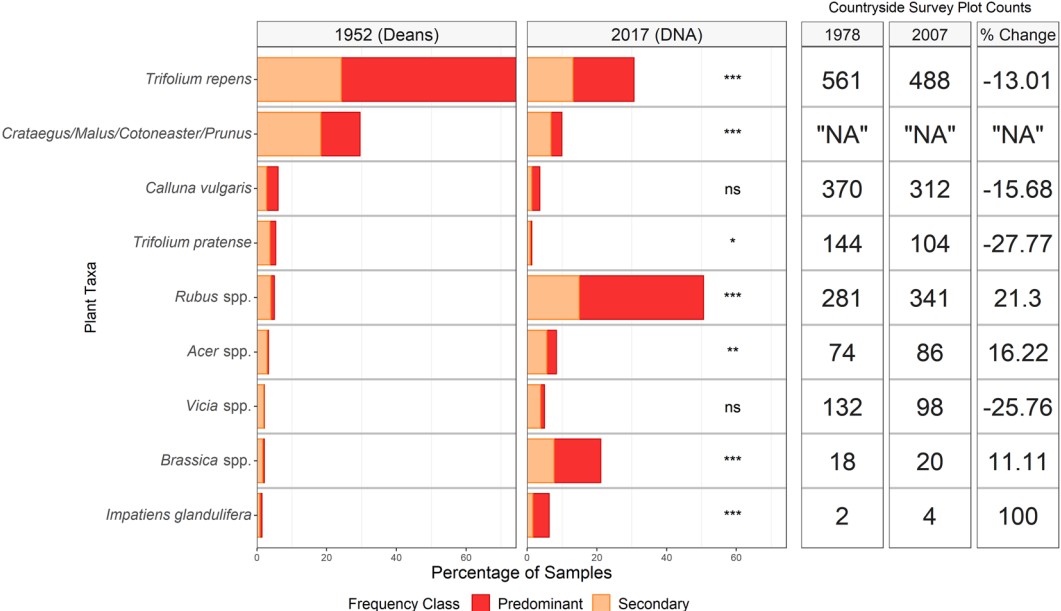

**Fig. 5 Change (%) in plant taxa used by honeybees from 1952 to 2017 along with changes in the abundance of those taxa in the Countryside Survey (%).** The taxa included are those found as predominant and secondary within honey samples (>15% of pollen grains in mellisopalynology or >15% of DNA sequences) for more than 1% of samples in both surveys. The p-value for the chi-squared tests used Bonferroni correction for multiple testing. The Countryside Survey represents changes in the abundance of the taxa within 1577 fixed plots between 1978 and 2007.

context of the wider pollinator community when discussing increasing forage provision. Honeybee foraging differs in comparison to other wild pollinators including solitary bees, bumblebees, and hoverflies who are not able to reach resources at the same scale. Honeybees' larger forage range comes from their large social structure coupled with their ability to communicate forage location[31]. In addition to behavioural differences, honeybees have physiological differences to other pollinators, including aspects such as tongue length, which can restrict access to other important sources of forage[32]. Recommendations for specific forage plants should therefore consider the needs of the wider pollinator cohort and the appropriate diversity of flowers required to meet the needs of a diverse and resilient pollinator community.

Here we show a significant correlation in the overall presence and absence of plant taxa found by both the DNA metabarcoding in 2017 and by melissopalynology in 1952, with significant differences when predominant and secondary foraging is examined. The robustness of DNA metabarcoding in accurately representing the abundance of biomass within a system is debated, and while positive correlations between the proportion of DNA sequences and relative abundance have been found, this can be accompanied by a high degree of variance between markers[33–36].

To our knowledge, this study represents the first UK-wide floral analysis of honey samples since the melissopalynology survey of 1952. The changes observed in honeybee foraging between 1952 and 2017 are evidencing the impacts of widespread changes in available forage from shifts in agricultural management, the presence of crop species and the spread of invasive species. Charting these changes has management implications for providing nectar and pollen forage nationally.

## Methods

**2017 honey sampling**. Beekeepers were invited to provide honey for analysis via a nationwide campaign publicised on the gardening programme, BBC Gardener's World (broadcast July 2017). Participating beekeepers were asked to supply ~30 ml of honey from any date in 2017, reporting the date of sample collection and the location of the apiary, using a grid reference or postcode. In total 441 honey samples were processed from beekeepers.

**Honey DNA extraction**. Any wax was removed using sterile forceps and DNA was extracted from 10 g of honey using a modified version of the DNeasy Plant Mini extraction kit (Qiagen). Firstly, the 10 g of honey was made up to 30 ml with molecular grade water and incubated in a water bath at 65 °C for 30 min. Samples were then centrifuged (Sorvall RC-5B) for 30 min at 15,000 rpm, the supernatant was discarded, and the pellet resuspended in 400 μL of a buffer made from a mix of 400 μL AP1 from the DNeasy Plant Mini Kit (Qiagen), 80 μL proteinase K (1 mg/ml) (Sigma) and 1 μL RNase A (Qiagen). This was incubated again for 60 min at 65 °C in a water bath and then disrupted using a TissueLyser II (Qiagen) for 4 min at 30 Hz with 3 mm tungsten carbide beads. The remaining steps were carried out according to the manufacturer's protocol, excluding the use of the QIAshredder and the second wash stage. The extracted DNA was purified using the OneStep PCR Inhibitor Removal Kit (Zymo Research) and diluted 1 in 10.

**PCR and library preparation**. Illumina MiSeq paired-end indexed amplicon libraries were created via a two-step PCR protocol. Two libraries were prepared for the DNA barcode regions, *rbcL* and ITS2. Initial amplification used the template specific primers *rbcL*af and *rbcL*r506[37], and ITS2F and ITS3R, with universal tails designed to attach custom indices in the second-round PCR. To improve clustering on the Illumina MiSeq, a 6N sequence was also added between the forward template specific primer and the universal tail.

Forward universal tail, 6N sequence and *rbcL*af: [**ACACTCTTTCCCTACACGACGCTCTTCCGATCT**]NNNNNN[ATGTCACCACAAACAGAGACTAAAGC]

Reverse universal tail and *rbcL*r506: [**GTGACTGGAGTTCAGACGTGTGCTCTTCCGATCT**][AGGGGACGACCATACTTGTTCA]

Forward universal tail, 6N sequence and ITS2F: [**ACACTCTTTCCCTACACGACGCTCTTCCGATCT**]NNNNNN[ATGCGATACTTGGTGTGAAT]

Reverse universal tail and ITS3R: [**GTGACTGGAGTTCAGACGTGTGCTCTTCCGATCT**][GACGCTTCTCCAGACTACAAT]

This first PCR used a final volume of 20 μl: 2 μl template DNA, 10 μl of 2× Phusion Hot Start II High-Fidelity Mastermix (New England Biolabs UK), 0.4 μl (2.5 μM) forward and reverse primers, and 7.2 μl of PCR grade water. Thermal cycling conditions for *rbcL* were: 98 °C for 30 s, 95 °C for 2 min; 95 °C for 30 s, 50 °C for 30 s, 72 °C for 40 s (40 cycles); 72 °C for 5 min, 30 °C for 10 s. Thermal cycling conditions for the first ITS2 PCR were: 98 °C for 30 s 94 °C for 5 min; 94 °C for 30 s, 56 °C for 30 s, 72 °C for 40 s (40 cycles); 72 °C for 10 min, 30 °C for 1 min. The initial PCR was carried out three times and pooled.

The pooled products from the first PCR were purified following Illumina's 16S Metagenomic Sequencing Library Preparation protocol using Agencourt AMPure XP beads (Beckman Coulter). The purified PCR product from round one was followed by a second round of amplification to anneal custom unique and identical i5 and i7 indices to each sample (Ultramer, Integrated DNA Technologies).

This index PCR stage used a final volume of 25 μl reaction (12.5 μl of 2× Phusion Hot Start II High-Fidelity Mastermix, 1 μl of i7 Index Primer and i5 Index Primer, 6.5 μl of PCR grade water, and 5 μl of purified first-round PCR product). Thermal cycling conditions were: 98 °C for 30 s; 95 °C for 30 s, 55 °C for 30 s, 72 °C

for 30 s (8 cycles); 72 °C for 5 min, 4 °C for 10 min. Following the index PCR, a 1% gel was run to verify its success. The index PCR product was then purified following the PCR clean-up two sections of the Illumina protocol. The purified products of the index PCR were quantified using a Qubit 3.0 fluorescence spectrophotometer (Thermo Fisher Scientific) and pooled at equal concentrations to produce the final library. Positive and negative controls were amplified and sequenced alongside honey samples. The positive control was made from a mixture of five tropical tree species that were not present in the survey site. The species *Baccaurea stipulata*, *Colona serratifolia.*, *Dillenia excelsa*, *Kleinhovia hospita*, and *Pterospermum macrocarpum* were used, taking 5 µl from each separate DNA extraction and mixing, before following the protocol as with the honey samples. All five species were detected within the sequencing results.

**Bioinformatic analysis**. Sequence data were processed using a modified data analysis pipeline[14,38]. Raw reads were trimmed to remove low-quality regions (Trimmomatic v. 0.33), paired, and then merged (FLASH v. 1.2.11), with merged reads shorter than 450 bp discarded. Identical reads were dereplicated within samples and then clustered at 100% identity across all samples (vsearch v. 2.3.2), with singletons (sequence reads that occurred only once across all samples) discarded.

The Barcode Wales and Barcode UK projects provide 98% coverage for the native flowering plants and conifers of the UK[37]. This reference library was supplemented with a curated library of the non-native and horticultural species, downloaded from GenBank. This UK species list was generated using the list of native species of the UK from Stace (2010)[39], 505 naturalised alien species (BSBI), and horticultural species from the IRIS BG database at the National Botanic Garden of Wales.

The sequence data from the honey samples were compared against the reference database using blastn, using the script vsearch-pipe.py. The top BLAST hits were then summarised using the script vsearch_blast_summary.py. Sequences with bit scores below the 1st percentile were excluded. If the top bit scores of a sequence matched to a single species, then the sequence was identified to that species. If the top bit scores matched to different species within the same genus, then the result was attributed to the genus level. If the top bit score belonged to multiple genera within the same family then a family level designation was made. Sequences that returned families from different clades were excluded. These automated identifications were then checked manually for botanical veracity. To check identified plant species against their availability across the UK, species records from the BSBI (Botanical Society of Britain and Ireland) were used for native species, while commercial availability for horticultural species was verified with the RHS Plant Finder[40]. Within each sample, the number of sequences returned from *rbcL* and ITS2 for each plant taxon was summed to combine the results of each marker.

The proportion of sequences was used in the analysis, which has been shown to be an appropriate method to control for differences in read number[41]. Alternatively, the sequencing data can be rarefied, but this has been criticised as a statistical technique, due to requiring the removal of valid data[41]. To investigate the impact of rarefying on the conclusions drawn from the data, all analyses were rerun with rarefied data (Supplementary Results).

**1952 Honey sampling**. In 1952, 855 honey samples were characterised from 66 counties across the UK and Ireland using melissopalynology[15,16]. The methods reported for the research conducted in 1952 are described here fully for comparison. Samples were obtained via a general appeal and were all collected during the honey season of 1952. For each honey sample, ~200 pollen grains were identified using the morphology of the pollen under the microscope, following a standardised protocol[42]. To extract the pollen, 10 g of honey was dissolved in 20 ml of distilled water, from which 10 ml was taken and centrifuged at ~2000 rpm for one minute. The supernatant was discarded, and the sediment retained, and then the process was repeated for the remaining liquid. From the sediment, a drop was transferred to a glass slide and spread out over an area of 1 cm$^2$, before being stained with fuchsin and dried. Euparal vert was used as a final mounting medium. Pollen was identified by comparison with a reference library of pollen preparations and available pollen morphological data[43,44]. Each plant taxon found in the sampled honey was reported according to the proportion of pollen grains found and classed into predominant (>45% of pollen grains), secondary (15–45% of pollen grains) and important minor (1–15% of pollen grains). The location data for the honey samples were restricted to the county level, and summary data tables were presented for each UK county that returned honey.

**Comparing the 1952 and 2017 honey samples**. The plants detected using DNA metabarcoding and melissopalynology have been compared in previous studies with concordance found between the two methods[45–48]. Both methods detect the same major taxa, but rarer species in a sample are less likely to be found consistently, both when comparing methods and also during replicates of the same method[45–47]. DNA metabarcoding is often able to detect more taxa when compared to melissopalynology, by identifying rarer species in the sample and by achieving higher taxonomic resolution in certain cases. While melissopalynology uses counts of pollen grains to provide a starting point for quantitative analysis, DNA metabarcoding as a process is semi-quantitative, with biases associated with

the process of DNA extraction, PCR and sequencing[33,45]. To allow for these considerations we placed the proportion of DNA sequence reads and pollen counts into four broad abundance classes matching the classifications used in melissopalynology (predominant, secondary, important minor and minor) and focus our analyses and conclusions on changes in the frequency of occurrence of the major taxa, classed as predominant and secondary. Both methods capture information on both nectar and pollen plants within the honey, however, certain species can be over or under represented in pollen analysis compared to their relative nectar contribution[49]. Both pollen and nectar plants are required to meet the foraging requirements of pollinators.

**Statistics and reproducibility**
*Statistical analysis of DNA metabarcoding data*. To understand how the plant taxa composition within the honey sample was structured in space and time, the effect of time (measured as the calendar month number in 2017), latitude and longitude of sampling location were included in a single, two-tailed generalized linear model using the 'manyglm' function in the package 'mvabund'[50]. Honey samples with missing metadata were excluded, giving a sample size of 428. An abundance table of taxa (number of sequence reads) found in each sample was set as the multi-variate response variable and a common set of predictor variables (month, latitude and longitude) were fit using a negative binomial distribution. The number of sequence reads per sample was included as an "offset" in the model in order to control for differences in the number of sequence reads between samples. Monte Carlo resampling was used to test for significant community-level responses to our predictors. The strong mean-variance relationship in the data (Supplementary Fig. 6) and the distribution of the count data (Supplementary Figs. 7, 8) support the use of a negative binomial distribution in the model. The appropriateness of the models was checked by visual inspection of the residuals against predicted values from the models (Supplementary Figs. 9–11).

We completed a spatial eigenfunction analysis using distance-based Moran's eigenvectors. Moran's Eigenvector Maps were computed using the 'mem' function from the adespatial package. Moran's I was computed for each taxa using the 'moran.randtest', with Bonferroni correction for multiple testing. The direction of autocorrelation (positive and negative) was tested using the 'moranNP.randtest' function, using the adespatial package in R.

*Statistical analysis of the 1952 and 2017 honey samples*. Abundance classes were assigned based on the percentage of reads returned for the two DNA regions *rbcL* and ITS2, matching the classifications used in melissopalynology. Plant taxa represented by over 45% of reads were designated *predominant* for that sample; between 15 and 45% were *secondary*; between 1 and 15% were *important minor* taxa, and <1% of reads were classed as *minor* taxa. The number of times each taxon occurred at each level of abundance was then calculated, with the sum of this giving the frequency of occurrence across all the samples.

The results of the 2017 analysis were then compared with 855 honey samples characterised in 1952, from across the UK and Ireland using melissopalynology[15,16]. The relationship between the frequency of occurrence for the matched plant taxa between 1952 and 2017 was assessed using Kendall's rank correlation.

To compare the major taxa (classed as predominant and secondary) between 2017 and 1952, the effect of sample location (UK county name) and sample year (2017 or 1952) were included in a two-tailed generalized linear model using the 'manyglm' function in the package 'mvabund'[50]. In the absence of latitude and longitude for honey samples collected in 1952, UK ceremonial county names were used as a proxy for the location for both 2017 and 1952 honey samples. Honey samples from 2017 were assigned their ceremonial county based on latitude and longitude and matched to the counties listed in 1952. Using a binomial distribution, the effect of county location and year (1952 and 2017) were included as explanatory variables in the model and the presence or absence of each taxa was set as the response variable. The appropriateness of the models was checked by visual inspection of the residuals against predicted values from the models (Supplementary Fig. 11).

The change in proportion of major (predominant and secondary) taxa between 1952 and 2017 was examined for the plant taxa that occurred as major taxa in more than 1% of samples for both honey surveys. Chi-squared contingency tests were used to assess differences, with Bonferroni correction for multiple testing. All statistical analyses were carried out using R (v. 3.5.2).

*Countryside Survey vegetation plot data frequency changes*. Changes in the local frequency of the major plant forage species found in both 1952 and 2017 were assessed using the Countryside Survey data from 1978 to 2007[51–53]. In 1978, the survey looked at 256 1 km squares within which fixed plots were established, representing fields and unenclosed land (200 m$^2$) as well as linear features including hedgerows, streams and roadsides (10 m$^2$). In each plot, a list of all vascular plants was recorded. Where possible, squares and plots were then revisited in 2007, representing 236 1 km squares containing 1577 plots. For these revisited plots, the percentage change in plot frequency was calculated[54].

*Landscape data*. The Land Cover 2017 map was used to characterise habitat in a 2 km radius of the hives[55] while the 2017 CEH Land Cover Plus: Crops map was

used to assess the presence and absence of crop species, *Brassica napus* (oilseed rape) and *Vicia faba* (field beans), within a 2 km radius of each hive. A chi-squared contingency test was used to analyse the differences between the presence of the crop species in the honey and the presence and absence of the crop within the landscape. Non-metric multidimensional scaling (NMDS) ordination was used to visualise differences in the composition of the honey relating to the dominant habitat type in a 2 km radius, based on the proportion of reads returned for each taxon. Ordinations were carried out using the metaMDS function in the vegan package[56] in R using Bray-Curtis dissimilarity indices. The differences in plant community composition and surrounding dominant habitat type were tested using the *adonis* function from vegan, with 999 permutations. Analyses and maps were generated in *R* (v. 3.5.2).

**Reporting summary**. Further information on research design is available in the Nature Research Reporting Summary linked to this article.

## Data availability

Sequence data are available at the Sequence Read Archive (SRA) accession number PRJNA577454. Summarised sequence data are provided in Supplementary Data.

## Code availability

The code for processing the sequencing data is available at https://github.com/colford/nbgw-plant-illumina-pipeline[38].

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

## Acknowledgements
Funding to support this research was provided by the National Botanic Garden of Wales and a bequest from Dr Quentin Kay. L.J. and A.L. were supported by Knowledge Economy Skills Scholarships (KESS2), part funded by the Welsh Government's European Social Fund (ESF). Thank you to beekeepers throughout the UK for generously providing honey samples. Thank you to Adam Leitch for making A.S.C. Deans' 1958 NDB thesis available to us.

## Author contributions
The study was conceived by NdV and L.J. The lab work was carried out by L.J. The data were compiled by L.J. and A.L. and analysed by L.J., C.F. and G.B. with suggestions from N.dV. and S.C. The manuscript was written by L.J. and N.dV. with contributions from all of the authors.

## Competing interests
The authors declare no competing interests.
