## [Peer Review File · Communications Biology]

Reviewers' comments:

Reviewer #1 (Remarks to the Author):

In this work, by de Vere and colleagues, the authors discuss an intriguing dataset which, when compared to a past analysis, appears to show changes in honey bee floral resource use across the UK between 1952 and 2017. In addition to the historical study, a nationwide vegetation survey is used to corroborate the findings. The manuscript is well-written, and the question posed and methods used to test it are intriguing. The key conclusions are that *Trifolium* species have decreased in abundance while bramble, cultivated Brassica and invasive Himalayan Balsam have increased in abundance. The authors also recommend that land managers increase *Trifolium* abundances in order to increase floral resource levels. The overall narrative is highly plausible in light of what is known about recent land use change. I do have several concerns about the methods and interpretation, however, which are detailed below.

Major concerns

1. This study examines honey samples from *Apis mellifera* and then generalizes the results to the broader pollinator community. I certainly see the appeal of honey bees in terms of accessibility and ease of study, however, there isn't enough discussion of the fact that what's best for the honey bee may not be best for the broader community. While there are challenges in honey bee health, the species is not of conservation concern (except for the local genetics of some areas of their native range). I don't think that makes honey bees useless for this type of application, but it certainly warrants careful discussion.
2. In performing some correlations between the ITS2 and rbcL proportions of each sample from the supplemental data provided, the precision of the methods is concerning. The two markers used provide quite different answers with respect to the contents of the samples. This raises a number of concerns, the foremost being which marker is providing the more accurate data? If neither of them is more optimal, is this enough data for the semi-quantitative type of inference being performed? I see the authors point that binning the results into the four categories potentially alleviates some of this error, but the imprecision is considerable, making it difficult to discern how robust the binning procedure is.
3. L234-235 detail the methods for pollen extraction from honey. How do these methods compare to those of Deans? I was unable to find an online copy of Deans work and, since this work is based largely on that study, it may be important for reviewers to have a chance to read it. This is particularly important in light of the many sampling factors that can taxonomically bias honey pollen analysis (e.g. <https://www.tandfonline.com/doi/pdf/10.1080/00173130410019497>)
4. As the paper is currently written, the authors are assuming a 1 to 1 relationship between honey pollen concentrations/detection frequencies and inferred nectar sources. There is clear consensus within the research community that nectar gathered from different types of flowers can have systematically different concentrations of pollen, often by orders of magnitude (see <https://doi.org/10.1080/01916122.2001.9989554>).
5. The methods are a bit unclear on how data from the two markers were jointly analyzed. From the supplementals, it seems clear that the raw read counts were summed across the markers for each sample, and then proportions were derived from those summed values. This is inappropriate in that libraries were sequenced with variable coverage. On average, ITS2 libraries were sequenced to a depth of ~9200 while rbcL libraries were sequenced to an average depth of ~13000. This differential coverage between ITS2 and rbcL means that the final estimates used for analysis are systematically weighted toward the rbcL results as currently analyzed. To correct this, the authors could randomly sample a standard number of reads across the markers.
6. L144-145 and throughout, the authors recommend that land managers increase the abundance of clovers within UK grasslands. Is there any data on how this may affect other flora/fauna within these

ecosystems? As I'm understanding, the suggestion is that we should increase the abundance of a single genus within remaining grassland habitat in order to compensate for lost habitat?

7. L315-316 states that "these automated identifications were then checked manually for botanical veracity." Could the authors provide more detail on what is meant by this?

Other concerns

1. L104, is the assumption here that because it's the most frequently used it is the most important? Pollinator nutrition is a very complicated multi-variate problem and I wonder if this oversimplifies the complexities involved.

2. L111-114, again, nutrition is very complicated, and I don't know that we can infer that *Rubus* is less-optimal just because it's lower in protein. In fact, in bumble bees, there is convincing evidence that the ratio of protein to lipids is more predictive of health/fitness effects (See work by Anthony Vaudo and Christina Grozinger). This field of research is still developing and I'm uncertain as to how well reference 14 substantiates the claim being made. In fact, reference 14 appears to be concerned exclusively with bumble bees.

3. L246, add references for rbCL primers.

4. L303-304, is this reference library provided?

5. L308-310, provide the minimum alignment percent IDs allowed in this pipeline. Did it differ by marker?

6. L328-330, with respect to NB GLM analysis, it seems like it would have been more appropriate to just use the raw count data in a mixed-model context. Scaling by 1000 and rounding to the nearest integer just serves to approximate the original count data without accounting for variance in sampling depth. A GLMM could account for this more appropriately.

Reviewer #2 (Remarks to the Author):

I have carefully read the entitled MS 'Shifts in honeybee foraging reveal historical changes in floral resources' by Jones et al. The originality of the paper lies in the comparison between historical nationwide survey of honey pollen composition samples in the UK (1952) with contemporary one (2017). This paper is thus in line with the influential papers dealing with historical data necessary to understand the decline of pollinators. The authors describe current plant composition used as foraged plants by honeybees and show shifts in honeybee foraging ; their results are in accordance with known changes in vegetation in the UK. The paper is well written and the figures are nice. My major concerns are:

1- More information are needed regarding to the 1952 nationwide survey (seasonality of the survey, location of the hives representative of historical land cover) to ensure that the foraged plants in 1952 are representative of historical floral resources availability.

2- The authors used the terms of changes in frequency of occurrence, abundance, frequency of major plants which is sometimes confusing. I would advise to explicit these terms early in the MS and to include some explanations about quantitative comparisons between DNA metabarcoding and melissopalynology in the methods (rather than in supplementary discussion).

3- I would advise to keep in mind that the actual use of floral resources is analysed through the Honeybee foraging, which may not be representative of all pollinators.

Below are other specific comments:

L. 31-32 and L.38-40. I am not totally convinced by the term 'foraging behaviour' because it suggests that the study involves 'behavioral work' which is not really the case. According to me, the strength of this study lies in the fact that the data enable the authors to assess how landscape changes impact the floral resources actually consumed by the honeybees (actual resources) and not the floral resources available to bees (potential resources).

L.33 please replace species by 'model'

L.36. Does pollen found within honey informs about pollen only or both nectar and pollen as stated?

L.48 'The types of habitat surrounding hives reflected the composition of habitats of the UK'. That sounds appropriate according to the Figure 1 however no statistical tests are provided. In particular, suburban and broadleaf woodlands appear to be overrepresented in the apiary. Does this impact the findings?

L. 54-55, L. 60 and thereafter: the authors make the differences between the 'frequent' and 'abundant' plant species. Some analyses deal with the occurrence of the species whereas the others deal with the abundance. I think this it is important to specify precisely what these terms refer to and why it is interesting to analyze one or the other or both of them according the sections.

L.66 'reflects the changes in the plants available to the honeybees, with calendar month'. Please rephrase including 'seasonal changes' to avoid any misleading with long term temporal changes presented hereafter.

L. 69-71. Why is it interesting to compare the results according to the Regions? Is it because the 1952 survey deals with honey collected in distinct regions?

L. 77. There are much less information provided regarding the 1952 honey sample collection compared to the 2017 one. In particular, I think it is important to mention the period of collection (does it match with the 2017 ones?), the location of the samples (is it possible to map the points even if it is a county reference?), and does the honey sample collection reflects the habitats founds around the 1950s? In other words, I would have been interested in seeing the Fig 1 and Fig2 for the 1952's data.

L. 80-81 please add 'the frequency of these 47 plant groups at the 2 collection dates (1952 vs 2017)' or something like that.

L. 97-99. One other explanation I see regarding *T. repens* changes is that the species may not have the ability to flower due to intensive grazing. This can explain why *Trifolium repense* may potentially offer higher potential floral resources than what is actually used.

L.103-108 Is there an underlined interpretation to explain this increase in bramble (in addition to the opposite trend with *T. repens*)?

L. 142-147 *Trifolium repens* from Improved grasslands appear as a good candidate to improve significantly the quantity of floral resources in the landscape for pollinators. However, keeping in mind that 1- not only the quantity but also the diversity of floral resources are necessary to maintain pollinators and 2-the actual use of resources is analyzed here through the honeybee foraging which is not representative of all pollinating insects, I would suggest to complete this management suggestion.

L. 317-322 As suggested before, more information about 1952 honey sampling would be useful to make sure the two periods are comparable (see my comment L. 77). Moreover, I am curious to know in which framework and for which goal this historical honey sampling have been conducted. Also, the 1952 honey samples came from the UK and Ireland whereas the 2017 mainly came from England, Scotland, Wales. How to ensure differences in honey pollen composition do not come from geographical differences? Perhaps, sensitivity analyses on a subset of the dataset (shared location between 1952 and 2017) would be interesting. Then, I am wondering why the authors did not include a proper section about 2017 honey collection in the methods.

L. 330 'The proportion of sequences was set as a response variable', but I am wondering if this proportion of sequences reflects well the abundance of the plant taxa in the samples. Is it reliable to pollen quantity of each species despite potential bias in sampling or PCR according to the species?

L. 339-346. The frequency classes were determined here on the basis of quantitative reads for 2017 samples. Please specify what is the quantitative measure for 1952 (the proportion of pollen grains I guess). Are they comparable?

L. 368-373 Changes in frequency of major plants were analysed according to the Countryside Survey dataset which provide "In each plot, a list of all vascular plants was recorded". I am wondering if it is a proxy of the abundance of plant taxa (as stated L. 88 for instance 'This corresponds to differences in their abundance') or rather a proxy of changes in frequency?) and why the %cover for each plant in

each plot were not analysed. Also, are only the shared plots between 1978 and 2007 surveys were included in the analysis?

Figure 2. Please add in '2017' in the title. Also, why the inset focus on the 'most frequent taxa found at predominant and secondary levels' and not simply the most frequent taxa?

Figure 3. In the maps, the colours of the dots are difficult to discriminate (red/orange). Moreover, the black dots are smaller than the orange/red ones which makes them difficult to distinguish. Please change only the colours, not the dot sizes. The legend of the below panels is not complete (it is not written what is specifically represented, colour classification is missing). Also, I would recommend to include the sample sizes (n) of the samples for each category.

Extended Data Figure 2. The majority of the 47 taxa present changes in their frequency status (from Important minor in 1952 to minor in 2017). How to explain this trend? Is it linked to the fact that DNA metabarcoding is more sensitive to rare species within samples?

Extended Data Fig 4/5/6 : if possible, I would suggest to move them in Supplementary L.492-507. This section brings important arguments to support the comparative approach between melissopalynology and DNA metabarcoding. This question came to mind all along the reading of the MS, I would thus suggest including some of these information in the method sections if possible. In order to prevent bias in quantitative approaches, the authors analysed the frequency changes of major plants classified as broad categories which sounds to me reasonable but here again, this information deserve to be presented in the MS to understand the figures and the results.

Then, I am wondering if the authors analysed the pesticide residues in their 2017 honey samples, according to location or habitat for example.

Reviewer #3 (Remarks to the Author):

This paper titled "Shifts in honeybee foraging reveal historical changes in floral resources" analyses historical changes in floral composition based on honey samples. The authors obtained floral resources data using two different methods (DNA metabarcoding and melissopalynology) to detect historical changes.

The authors found a change in the use of major forage plants reflecting agricultural intensification, such as the increase of food crops and invasive species. Overall, the questions and methods used are interesting, and it is clear and well written. I also think that this paper has some merits because it uses historical data to evaluate ecological changes. I do not find, however, the results particularly surprising, but they show the importance of using honey samples to monitor floral resources.

However, I have some major concerns with this study mainly related to the validity of the methods and conclusions drawn from the analysis:

- Comparison between DNA metabarcoding and melissopalynology: Although I see the value of comparing two different data sets obtained from two different methods (and I understand that this is particularly difficult with historical data), I think there are few important limitations and biases that need to be tackled before drawing some relevant conclusions. The authors acknowledge these limitations in the "Supplementary Discussion" and try to overcome the main problem of doing a quantitative comparison between microscopy and DNA metabarcoding of honey samples by using four broad classes and focusing on the "abundant" (predominant and secondary) classes. This could be a valid approach, however the problem of taxonomic resolution persists. As the authors explain (lines 512-516) some taxa (19) were not identified but the authors do not know if this is because DNA metabarcoding (and may be its associated reference database) was not able to detect (potentially due to other biases (e.g. PCR biases)) or because they are actually not there in those sites. The level of uncertainty is high and hard to control given that there are many confounding factors (e.g. time, location, PCR biases, reference database). I will talk about the location problem specifically in my next

point because I have some suggestions to disentangle this effect (see below). Ideally, DNA metabarcoding would have been applied to both 1952 and 2017 samples and/or the study would have gone back to the same sites where 1952 honey samples were taken. Given the constraints of this study, it is hard to make major conclusions without other lines of evidence.

- Geographical/location biases: to understand the effect of space and time, as well as comparing 1952 vs. 2017, the authors did multiple GLMs taking time, latitude and longitude as covariates of the model. My first suggestion is to explicitly put the equations of the GLMs, this always helps to make clear what is the model being analysed besides doing it verbally, it avoids confusions. Secondly, did the authors check for potential spatial (and temporal) autocorrelation effects? Given that the authors use lat/lons as factors in their models this might violate the assumptions of independence, it might be important to check any spatial autocorrelation effects. For example, besides doing a Moran I test, the authors could plot the residuals versus the lat and lon variable (or even the sampling time of the 2017 honey samples). If there is auto-correlation, a potential solution for this could be to use location as a random effect. I suggest the authors following the regression protocol of Zuur & Ieno 2016 (Methods in Ecol and Evol, 7: 636-645). By doing a GLMM the authors might be able to disentangle some of the confounding factors.

Minor comments:

- Lines 87-88: Although I see the value of using the countryside survey data, there is a large temporal gap between 1952 and 1978. The data is good to indicate some changes of the flora across the UK, however the authors have to be cautious drawing conclusions about changes of 'abundance', especially when comparing with DNA metabarcoding data. I will focus more generally on changes in composition rather than 'abundances'.
- Lines 332-333: I think these transformations are a bit arbitrary; did the authors test other models (e.g. Normal or Gamma distribution) where you don't need to convert to integers the data?
- I suggest the authors to compare beta diversity indices between 1952 and 2017 (besides comparing broad class frequencies), and perhaps doing a multivariate analysis (e.g. PERMANOVA) or ordination analyses. These approaches are normally used for metabarcoding data (Laroche et al 2017, PeerJ 5: e3347)
- Figure 1: could the authors add a map of 1952 honey samples?
- Extended data Figure 5: could the authors add a figure of Pearson residuals versus other covariates (latitude, longitude, time).

Point-by-point response to the reviewers' comments and suggestions

Thank you ever so much to the three reviewers' who have provided their time and expertise reviewing our manuscript "Shifts in honeybee foraging reveal historical changes in floral resources". We really appreciate their careful and thorough analysis and have worked hard to fully incorporate all of their suggestions into this new, substantially revised, version of the manuscript. We have incorporated new statistical analysis and extended and reworked all sections of the MS to address the reviewers' comments. We have also modified our paper to ensure that it complies with the editorial and data policies of Communications Biology.

Reviewers' comments:

Reviewer #1 (Remarks to the Author):

In this work, by de Vere and colleagues, the authors discuss an intriguing dataset which, when compared to a past analysis, appears to show changes in honey bee floral resource use across the UK between 1952 and 2017. In addition to the historical study, a nationwide vegetation survey is used to corroborate the findings. The manuscript is well-written, and the question posed and methods used to test it are intriguing. The key conclusions are that *Trifolium* species have decreased in abundance while bramble, cultivated Brassica and invasive Himalayan Balsam have increased in abundance. The authors also recommend that land managers increase *Trifolium* abundances in order to increase floral resource levels. The overall narrative is highly plausible in light of what is known about recent land use change. I do have several concerns about the methods and interpretation, however, which are detailed below.

Major concerns

1. This study examines honey samples from *Apis mellifera* and then generalizes the results to the broader pollinator community. I certainly see the appeal of honey bees in terms of accessibility and ease of study, however, there isn't enough discussion of the fact that what's best for the honey bee may not be best for the broader community. While there are challenges in honey bee health, the species is not of conservation concern (except for the local genetics of some areas of their native range). I don't think that makes honey bees useless for this type of application, but it certainly warrants careful discussion.

- There isn't enough discussion of the fact that what's best for the honey bee may not be best for the broader community

We agree that the foraging behaviour of honeybees cannot represent the whole pollinator community. Honeybees do provide a good model for the study of landscape forage as their longer foraging distance can represent more of the wider landscape. Honey additionally gives us the longer-term foraging effort of multiple bees compared to sampling individual pollinators. However, their positives in terms of representing overall forage availability in the landscape can set them apart from other pollinators who have shorter average foraging distances and cannot communicate the source of forage between individuals. In addition, they have physical differences (such as tongue morphology) to other pollinators in terms of their ability to access nectar. To address these points we have added to the discussion starting at L242.

2. In performing some correlations between the ITS2 and *rbcL* proportions of each sample from the supplemental data provided, the precision of the methods is concerning. The two markers used provide quite different answers with respect to the contents of the samples. This raises a number of concerns, the foremost being which marker is providing the more accurate data? If neither of them is more optimal, is this enough data for the semi-quantitative type of inference being performed? I see the authors point that binning the results into the four categories potentially alleviates some of this error, but the imprecision is considerable, making it difficult to discern how robust the binning procedure is.

- Which marker is providing the more accurate data?

In this study two plant markers *rbcL* and ITS2 were used to assess the floral source of honey. Each marker alone will sample the plant communities within the honey differently, depending on the universality of the region. We know that *rbcL* provides the greatest sequence recovery (i.e. universality) for the plant flora of the UK, based on our UK DNA plant barcode database, while providing more limited species discrimination. ITS2 can provide increased species discrimination, however the ability to successfully amplify a barcode sequence from as many plant taxa as *rbcL* is lower, again something we see from our analyses of the UK DNA barcode reference library. This is illustrated in the below figure, which shows the ability to successfully recover a DNA barcode sequence for the UK flora (n = 1,482), and the discrimination ability of those sequences (Jones, 2019 PhD thesis, Jones et

Figure 1: The overall level of taxonomic representation in the database for the species of the UK flora. Recoverability (a) shows the level of representation for the native species of the UK flora (n = 1,482) in the reference library. Discrimination (b) shows the taxonomic resolution achieved using BLAST for those plant species in the reference library which were represented by all three markers more than once (n = 634).

al. in prep).

From the results of the honey samples here, we see a similar pattern with *rbcL* overall identifying a higher number of unique taxa (*rbcL* n = 121, ITS2 n = 84), but ITS2 can show some increased discrimination in the species it does detect e.g., ITS2 can distinguish the *Sambucus/Viburnum* grouping, while *rbcL* cannot. By using both *rbcL* and ITS2 we do not lose any information on the floral content of the honey and consider it more appropriate to

combine the data to retain all of the information provided by the two markers of the plant community. Consequently, after years of optimization, we choose to use a *rbcL* to amplify the widest breadth of taxa and ITS2 to augment taxonomic group resolution.

- Is this enough data for the semi-quantitative type of inference being performed?

DNA metabarcoding data is referred to as semi-quantitative, with biases potentially arising throughout the process of DNA extraction, PCR and sequencing. Using the read count to calculate the relative proportion of the taxa present in a sample is often used to provide a semi-quantitative measure of the relative biomass contribution in a system (Brennan 2019, Richardson 2018, Erikson 2017, Kartzinel 2015). Significant positive relationships have been found between pollen abundance or the amount of pollen DNA with the abundance of sequencing reads in experimental samples (Baksay et al., 2020). When looking at the pollen loads from honeybees, studies have found a positive relationship between abundance estimates using pollen counts and morphological identification and abundance of returned sequencing reads (Keller et al., 2015; Richardson et al., 2018; Smart et al., 2017). With this in mind, when analysing the 2017 honey samples and the relationship between plant composition and sampling time and location, we have confidence that it is appropriate to use the abundance information provided by the read counts.

Comparisons between DNA metabarcoding and melissopalynology methods found that the most abundantly found plant taxa were the most consistently detected between the two survey methods, whereas rare taxa were less likely to be detected by both surveys (Hawkins et al., 2015; Richardson et al., 2015). When comparing this data with the melissopalynology data, we use the frequency of occurrence of each taxon across samples and only compare frequency of occurrence changes in the taxa when they were found abundantly (>15% of sequencing reads or pollen grains), as we know the abundant taxa are more consistently found by both survey methods.

3. L234-235 detail the methods for pollen extraction from honey. How do these methods compare to those of Deans? I was unable to find an online copy of Deans work and, since this work is based largely on that study, it may be important for reviewers to have a chance to read it. This is particularly important in light of the many sampling factors that can taxonomically bias honey pollen analysis

(e.g. <https://www.tandfonline.com/doi/pdf/10.1080/00173130410019497>)

- How do the 2017 methods compare to those of Deans?
- This is particularly important in light of the many sampling factors that can taxonomically bias honey pollen analysis

Thank you for this reference, comparing between techniques for extracting pollen from honey, including the use of ethanol versus water and the length of time centrifuging. In both 1952 and 2017 water was used to dilute 10 g honey. In 2017 a higher speed centrifuge was used at 13,000 rpm for 30 min, while in 1952 the liquid was centrifuged for one min at 2000 rpm. The length of time (1 min versus 10 min) was not found to cause a significant difference in pollen concentrations or taxa found between the water diluted samples in Jones and Bryant (2004). To make these differences in technique clear, the 1952 methods have been expanded at L362 and the reference that Deans followed for the protocol in 1952 has been added (Maurizio & Hodges, 1951), which is more accessible.

4. As the paper is currently written, the authors are assuming a 1 to 1 relationship between honey pollen concentrations/detection frequencies and inferred nectar sources. There is clear consensus within the research community that nectar gathered from different types of flowers can have systematically different concentrations of pollen, often by orders of magnitude (see <https://doi.org/10.1080/01916122.2001.9989554>).

Thank you for this reference which discusses the research for scaling pollen melissopalynology data to provide a more accurate representation of the nectar contribution to the honey. This is highly important to the sale and regulation of honey as a commercial product and to understand the actual source of the nectar of the plants making up most of the honey. As forage sources for bee health, we are also interested in the plants that may provide limited nectar but ample pollen which is also important to the survival of the hive. How much pollen foraging is captured within the honey is another aspect which is difficult to separate out from nectar foraging as many plants can provide both nectar and pollen. This has been added as a point to the manuscript at L409.

5. The methods are a bit unclear on how data from the two markers were jointly analyzed. From the supplementals, it seems clear that the raw read counts were summed across the markers for each sample, and then proportions were derived from those summed values. This is inappropriate in that libraries were sequenced with variable coverage. On average, ITS2 libraries were sequenced to a depth of ~9200 while *rbcL* libraries were sequenced to an average depth of ~13000. This differential coverage between ITS2 and *rbcL* means that the final estimates used for analysis are systematically weighted toward the *rbcL* results as currently analyzed. To correct this, the authors could randomly sample a standard number of reads across the markers.

- The methods are a bit unclear on how data from the two markers were jointly analysed

The reviewer is correct to point out that the raw read counts from each marker were summed and proportions were calculated from the summed values. At L359, further description of this has been added in the methods to make this clearer within the manuscript.

- This differential coverage between ITS2 and *rbcL* means that the final estimates used for analysis are systematically weighted toward the *rbcL* results as currently analyzed.

The estimates of abundance will be on average weighted towards *rbcL* because of the greater read count for the *rbcL* libraries versus the ITS2. From the creation of the UK DNA barcode library, we know that *rbcL* is more successful in recovering sequences from the UK flora than ITS2 and therefore consider it our ‘backbone’ marker which is more likely to detect more of the plant taxa present within the honey (Jones, 2019 PhD thesis, Jones *et al.* in prep). We consider the weighting towards *rbcL* as a reflection of this increased recoverability and would not want to increase the impact of ITS2 given that it cannot detect some of the common honey plants, found using *rbcL* and melissopalynology e.g. *Salix*.

6. L144-145 and throughout, the authors recommend that land managers increase the abundance of clovers within UK grasslands. Is there any data on how this may affect other

flora/fauna within these ecosystems? As I'm understanding, the suggestion is that we should increase the abundance of a single genus within remaining grassland habitat in order to compensate for lost habitat?

- Is there any data on how this may affect other flora/fauna within these ecosystems?
- Is the suggestion that we should increase the abundance of a single genus within the remaining grassland habitat in order to compensate for lost habitat?

The recommendations for increased abundance and flowering are focused on improved grassland systems in the UK, which are typically intensively farmed and managed land. These intensively managed areas are generally species poor botanically and of limited ecological interest. The data here provides further encouragement for management recommendations to add *Trifolium* species into sown grass seed mixes and allow it to flower. Both red clover (*Trifolium pratense*) and white clover (*Trifolium repens*) can be sown into livestock grazed fields to provide a source of protein. They also fix nitrogen to help with reducing the requirement for artificial nitrogen fertiliser required for grass growth. However, heavy grazing or multiple early cuts can prevent the *Trifolium* from reaching flowering, so management should consider this in order to benefit pollinators. The discussion in the manuscript has been amended from L191 to reflect that we are discussing managed, reseeded landscapes when referring to improved grassland.

7. L315-316 states that “these automated identifications were then checked manually for botanical veracity.” Could the authors provide more detail on what is meant by this?

For each identification, this was verified against botanical information such as UK plant distribution and horticultural availability. The UK native and naturalised plant flora is well mapped with recording schemes by the BSBI (Botanical Society of Britain and Ireland) and data presented in the UK plant atlas, while the widespread commercial availability of horticultural plant species was judged using the RHS (Royal Horticultural Society) Plant Finder, which publishes a list of plants supplied by UK nurseries each year. This information was used to assess the likelihood of the automated identification, for example if a species level identification was to a species which was rare, and highly locally distributed then it was changed to a genera level identification. In most cases, the assigned identification was corrected to higher taxonomic level, e.g. from species to genus. This wording has been clarified in the methods section at L356.

In addition, our knowledge of the UK DNA barcode library provided context for the identification abilities of the two barcode regions. For example, *rbcL* cannot distinguish between the genera *Sambucus* and *Viburnum*, both likely honey forage plants, and the results were amended to reflect this.

Other concerns

1. L104, is the assumption here that because it's the most frequently used it is the most important? Pollinator nutrition is a very complicated multi-variate problem and I wonder if this oversimplifies the complexities involved.

L104: “Contrasting the decline in the *Trifolium* species, bramble (*Rubus fruticosus*) has seen an increase in forage use compared to 1952 and is now the most important forage plant for honeybees in the UK”

At L200 “most important forage plant” has been changed to the “most foraged plant” in order to prevent the over-simplification that the use of the word important gives.

2. L111-114, again, nutrition is very complicated, and I don’t know that we can infer that *Rubus* is less-optimal just because it’s lower in protein. In fact, in bumble bees, there is convincing evidence that the ratio of protein to lipids is more predictive of health/fitness effects (See work by Anthony Vaudo and Christina Grozinger). This field of research is still developing and I’m uncertain as to how well reference 14 substantiates the claim being made. In fact, reference 14 appears to be concerned exclusively with bumble bees.

L111-114: “. *R. fruticosus* and *T. repens* both offer pollen and nectar; however, the protein content and proportion of essential amino acids is lower in *R. fruticosus* compared to *T. repens* which has implications for honeybee health¹⁴.”

Within the paper referenced, the amino acids they considered essential for bumblebees were those which had been identified as essential for honeybees. They then characterised the pollen protein content and proportion of essential amino acids within bumblebee forage plants, some of which we also detect in the honey, including, *Trifolium repens* and *Rubus fruticosus*. The reviewer is correct to say that it is an oversimplification that because *Rubus* has less available protein and essential amino acids compared to *Trifolium* that it is less optimal. Given that this is an area of developing research the nutritional impact of decreased foraging on *Trifolium repens* and increased foraging on *Rubus fruitcosus* is unknown and is an interesting area for further research to explore. We have amended the discussion at L209 to reflect this and moved the location of the citation to more clearly indicate what was being referenced.

3. L246, add references for *rbcL* primers.

A reference has been added for the *rbcL* primers at L289.

4. L303-304, is this reference library provided?

The script to create the reference library has been archived on GitHub and is available at the link at L468. All of the Barcode UK data is available on GenBank and BOLD. GenBank accessions: JN890545-JN896265; KX165423-KX167996; MK924423-MK926404.

5. L308-310, provide the minimum alignment percent IDs allowed in this pipeline. Did it differ by marker?

There was not a minimum alignment percent ID cut-off for output in this pipeline. The checking of each sequence looked at the bitscore value returned for each match and moved it to a higher taxonomic assignment if the bitscore was poor.

6. L328-330, with respect to NB GLM analysis, it seems like it would have been more appropriate to just use the raw count data in a mixed-model context. Scaling by 1000 and rounding to the nearest integer just serves to approximate the original count data without

accounting for variance in sampling depth. A GLMM could account for this more appropriately.

Following reviewers' suggestions, we have now updated the model to include count data instead of proportional count data. We used proportion of each taxa (relative to the number of sequences in a sample) to control for differences in sequence read numbers from each sample (McMurdie et al 2014). However, we have now included the number of sequences from each sample as an "offset" in the manglm model (Supplementary Figure 10) which is included in the linear predictor during fitting and removes the need to scale the data. This modification has not changed the conclusions drawn from the model.

We have also updated the text for the statistical analysis to make it clear that the model was run a single time with a multivariate response (opposed to multiple glm models performed individually for each taxa). The manyglm modelling approach is therefore an excellent option for analysing multivariate abundance data produced by metabarcoding, within a single model. Please see changes to statistical analyses section for clarity from L377-393. Please also note that it is not possible to add random variables to the manyglm model (like a glmm).

Reviewer #2 (Remarks to the Author):

I have carefully read the entitled MS 'Shifts in honeybee foraging reveal historical changes in floral resources' by Jones et al. The originality of the paper lies in the comparison between historical nationwide survey of honey pollen composition samples in the UK (1952) with contemporary one (2017). This paper is thus in line with the influential papers dealing with historical data necessary to understand the decline of pollinators. The authors describe current plant composition used as foraged plants by honeybees and show shifts in honeybee foraging; their results are in accordance with known changes in vegetation in the UK. The paper is well written and the figures are nice. My major concerns are:

1- More information are needed regarding to the 1952 nationwide survey (seasonality of the survey, location of the hives representative of historical land cover) to ensure that the foraged plants in 1952 are representative of historical floral resources availability.

- seasonality of the survey,
- location of the hives representative of historical land cover

As with 2017, the honey was asked for in a general survey. For beekeeping in the UK, the most common time to take honey off the hives is in July and August. More detail has been added to the 1952 methods section at L361 to provide greater clarity on the survey. Location information on the hives is restricted to county level for the 1952 data. To check for the impact of the sampling location, this was included in the comparison between 1952 and 2017 and was not found to explain the differences in the data as explained in L148.

2- The authors used the terms of changes in frequency of occurrence, abundance, frequency of major plants which is sometimes confusing. I would advise to explicit these terms early in the MS and to include some explanations about quantitative comparisons between DNA metabarcoding and melissopalynology in the methods (rather than in supplementary discussion).

- explicit these terms early in the MS

Frequency of occurrence is the number of times a plant taxon is found across all of the honey samples. Abundance refers to the abundance classes the plant taxon was found at within a honey sample (predominant, secondary, important minor, and minor) which match the melissopalynology classes. We have added an explanatory section to make this clearer earlier in the MS at L101.

3- I would advise to keep in mind that the actual use of floral resources is analysed through the Honeybee foraging, which may not be representative of all pollinators.

Thank you for this comment and in light of a similar comment from Reviewer 1, we have added further discussion to highlight the application of honeybee foraging to wild pollinators at L242.

Below are other specific comments:

L. 31-32 and L.38-40. I am not totally convinced by the term ‘foraging behaviour’ because it suggests that the study involves ‘behavioral work’ which is not really the case. According to me, the strength of this study lies in the fact that the data enable the authors to assess how landscape changes impact the floral resources actually consumed by the honeybees (actual resources) and not the floral resources available to bees (potential resources).

Foraging behaviour has been changed to forage availability at L71 and L72.

L.33 please replace species by ‘model’

The word species has been changed to model at L71

L.36. Does pollen found within honey informs about pollen only or both nectar and pollen as stated?

- inform about pollen only, or both nectar and pollen?

Pollen found within the honey informs about both nectar and pollen plants, given that pollen can end up in the honey through several routes. One from pollen present in the nectar stored by the bee, two, from pollen carried on the body of the bee. Grooming that occurs in the hive near open cells can allow pollen to enter uncapped honey. In our analysis of spring foraging, we also completed pollen trapping at the same time as collecting honey, where the pollen baskets are taken from foraging bees. All of the plants that were found to be used by the pollen foraging bees were also present in the honey.

L.48 ‘The types of habitat surrounding hives reflected the composition of habitats of the UK’. That sounds appropriate according to the Figure 1 however no statistical tests are provided. In particular, suburban and broadleaf woodlands appear to be overrepresented in the apiary. Does this impact the findings?

- No statistical tests are provided

Thank you for noting this omission. A Spearman’s rank correlation was run to measure the association between the area of habitat surrounding the hives and of the UK ($r_s = 0.8$, $p =$

0.0002), demonstrating the reflection of habitats at the local and national levels. The results of this have been added to the manuscript at L94.

- In particular, suburban and broadleaf woodlands appear to be overrepresented in the apiary. Does this impact the findings?

During initial data exploration NMDS plots were used to look at whether the dominant habitat type surrounding the hive predicted the floral composition, but no pattern was observed in the data between the floral composition recorded in the honey and the surrounding dominant habitat type. We did not observe any clustering of suburban or urban dominant honey samples, or the broad-leaved woodland, as illustrated in the below plots. The habitat ordination plot has now been included in the supplementary information and a line added to the methods at L462, the results at L127.

Figure: NMDS plot of the all honey samples coloured by month, where we can see the clustering of the April and May samples to the left of the plot.

Figure: NMDS plot of the only the honey samples collected in July and August, coloured by dominant surrounding habitat.

L. 54-55, L. 60 and thereafter: the authors make the differences between the ‘frequent’ and ‘abundant’ plant species. Some analyses deal with the occurrence of the species whereas the others deal with the abundance. I think this it is important to specify precisely what these terms refer to and why it is interesting to analyze one or the other or both of them according the sections.

Frequency of occurrence is the number of times a plant taxon appears across all of the honey samples. Abundance refers to the percentage of sequence reads returned within a honey sample, and whether the plant taxon represented the majority of sequences returned within the honey sample, as designated by abundance classes. We then examined the frequency changes between honey samples where plant taxa were found abundantly. To make this explicit in the manuscript we have added a clarifying section at L98 and L101.

L.66 ‘reflects the changes in the plants available to the honeybees, with calendar month’. Please rephrase including ‘seasonal changes’ to avoid any misleading with long term temporal changes presented hereafter.

Rephrased to include ‘seasonal changes’ to add clarity at L122.

L. 69-71. Why is it interesting to compare the results according to the Regions? Is it because the 1952 survey deals with honey collected in distinct regions?

There are broad distinctions between the regions in terms of habitat which we were interested in investigating. Wales, compared to England, has less arable land and so has less available oilseed rape, (*Brassica napus*) or field beans (*Vicia faba*) for the bees to forage on. Scotland

is associated with heather and heather grassland, so we might expect more *Calluna vulgaris* within Scottish samples.

L. 77. There are much less information provided regarding the 1952 honey sample collection compared to the 2017 one. In particular, I think it is important to mention the period of collection (does it match with the 2017 ones?), the location of the samples (is it possible to map the points even if it is a county reference?), and does the honey sample collection reflects the habitats found around the 1950s? In other words, I would have been interested in seeing the Fig 1 and Fig2 for the 1952's data.

- Period of collection
- Location of the samples
- Does the honey sample collection reflect the habitats found around the 1950s?

To address this, further information on the sampling in 1952 has been added to the methods at L362. In both 1952 and 2017 honey was collected when beekeepers would naturally remove honey from the hives. The location information on the historical samples was restricted to county level. In order to provide a comparison between the county level sampling between 1952 and 2017, a heat map of the sampling locations in both years has been added as a supplementary figure (Supplementary Fig. 5).

L. 80-81 please add 'the frequency of these 47 plant groups at the 2 collection dates (1952 vs 2017)' or something like that.

At L143 "between the two collection dates of 1952 and 2017" was added to make it clearer.

L. 97-99. One other explanation I see regarding *T. repens* changes is that the species may not have the ability to flower due to intensive grazing. This can explain why *Trifolium repens* may potentially offer higher potential floral resources than what is actually used.

This is a very good point, which we have added to the discussion at L197.

L.103-108 Is there an underlined interpretation to explain this increase in bramble (in addition to the opposite trend with *T. repens*)?

The Countryside Survey data indicated an increase in frequency of bramble between the survey years, while the New Atlas of the British and Irish Flora (Preston et al. 2002) reported no change in the UK distribution of *Rubus fruticosus* agg., between 1962 and 2001. This has been incorporated into the discussion at L205.

L. 142-147 *Trifolium repens* from Improved grasslands appear as a good candidate to improve significantly the quantity of floral resources in the landscape for pollinators. However, keeping in mind that 1- not only the quantity but also the diversity of floral resources are necessary to maintain pollinators and 2-the actual use of resources is analyzed here through the honeybee foraging which is not representative of all pollinating insects, I would suggest to complete this management suggestion.

Further discussion at L242 has been added to reflect upon applying honeybee foraging to the wider pollinator community, and the management suggestion has been changed at L239 to incorporate plant diversity.

L. 317-322 As suggested before, more information about 1952 honey sampling would be useful to make sure the two periods are comparable (see my comment L. 77). Moreover, I am curious to know in which framework and for which goal this historical honey sampling have been conducted. Also, the 1952 honey samples came from the UK and Ireland whereas the 2017 mainly came from England, Scotland, Wales. How to ensure differences in honey pollen composition do not come from geographical differences? Perhaps, sensitivity analyses on a subset of the dataset (shared location between 1952 and 2017) would be interesting. Then, I am wondering why the authors did not include a proper section about 2017 honey collection in the methods.

- In which framework and for which goal this historical honey sampling was conducted

A.S.C Deans survey of UK honey was research conducted as a thesis for the National Diploma in Beekeeping and submitted in 1958. The diploma represents the highest beekeeping qualification in the UK. The survey was the first large-scale survey of UK honey sources. Within Deans' thesis, he presents summary tables for each UK county sampled, giving the number of samples and each plant taxon found. A shorter summary report was published in 1957 by the Bee Research Association.

- How to ensure differences in honey pollen composition do not come from geographical differences?

The effect of sample county location was included as an explanatory variable in the model comparing between the plants found in 1952 and 2017 and was not found to significantly explain the differences in the plants found within the honey.

- Then, I am wondering why the authors did not include a proper section about 2017 honey collection in the methods.

The 2017 honey collection methods were outlined under honey sampling and DNA extraction in the methods. An appeal for honey samples was publicised for honey extracted in 2017 from one hive, supplying approximately 30 ml in a sterile tube. The method headings have been changed at L266 to make this clearer.

L. 330 'The proportion of sequences was set as a response variable', but I am wondering if this proportion of sequences reflects well the abundance of the plant taxa in the samples. Is it reliable to pollen quantity of each species despite potential bias in sampling or PCR according to the species?

DNA metabarcoding data is referred to as semi-quantitative, with biases potentially arising through the process of DNA extraction, PCR and sequencing. Using the read count to calculate the relative proportion of the taxa present in a sample is often used to provide a semi-quantitative measure of the relative biomass contribution in a system (Brennan 2019, Richardson 2018, Erikson 2017, Kartzinel 2015). Significant positive relationships have been found between pollen abundance or the amount of pollen DNA with the abundance of sequencing reads in experimental samples (Baksay et al., 2020). When looking at the pollen loads from honeybees, studies have found a positive relationship between abundance estimates using pollen counts and morphological identification and abundance of returned

sequencing reads (Keller et al., 2015; Richardson et al., 2018; Smart et al., 2017). For honey samples, comparisons between DNA metabarcoding and melissopalynology found that the most abundantly found plant taxa were the most consistently detected between the two survey methods, whereas rare taxa were less likely to be detected by both surveys (Hawkins et al., 2015; Richardson et al., 2015).

With this in mind, when analysing the 2017 honey samples and the relationship between plant composition and sampling time and location, we think it appropriate to use the abundance information provided by the read counts. However, when comparing this data with the melissopalynology data, we use the frequency of occurrence of each taxon across samples and only compare frequency of occurrence changes in the taxa when they were found abundantly (>15% of sequencing reads or pollen grains), as we know the abundant taxa are more consistently found by both survey methods.

L. 339-346. The frequency classes were determined here on the basis of quantitative reads for 2017 samples. Please specify what is the quantitative measure for 1952 (the proportion of pollen grains I guess). Are they comparable?

The proportion of pollen grains was the basis for the quantitative measure in 1952. The methods have been expanded at L361 to make this clearer within the manuscript.

L. 368-373 Changes in frequency of major plants were analysed according to the Countryside Survey dataset which provide “In each plot, a list of all vascular plants was recorded”. I am wondering if it is a proxy of the abundance of plant taxa (as stated L. 88 for instance ‘This corresponds to differences in their abundance’) or rather a proxy of changes in frequency?) and why the %cover for each plant in each plot were not analysed. Also, are only the shared plots between 1978 and 2007 surveys were included in the analysis?

The “differences in their abundance” has been changed to difference in their frequency at L152 as the Countryside Survey dataset indicates. Only the shared plots between 1978 and 2007 are included in the analysis following the methods outlined by Carvell (2006).

Figure 2. Please add in ‘2017’ in the title. Also, why the inset focus on the ‘most frequent taxa found at predominant and secondary levels’ and not simply the most frequent taxa?

2017 has been added to the figure title for Fig. 2. The decision to focus the inset on taxa found at the predominant and secondary abundance levels allows us to highlight the plant taxa which represent major forage (predominant or secondary) but were found at lower frequency across all of the honey samples due to their flowering phenology being at the start or end of the season such as the spring *Taraxacum* or the autumn flowering *Impatiens glandulifera*, and *Calluna vulgaris*.

Figure 3. In the maps, the colours of the dots are difficult to discriminate (red/orange). Moreover, the black dots are smaller than the orange/red ones which makes them difficult to distinguish. Please change only the colours, not the dot sizes. The legend of the below panels is not complete (it is not written what is specifically represented, colour classification is missing). Also, I would recommend to include the sample sizes (n) of the samples for each category.

The orange/red dots have been changed to black triangles to make them easier to distinguish, and the caption changed to reflect the figure. The below table has been added as a supplementary table to provide the sample sizes (Supplementary Table 1).

Abundance Class (Percentage of sequencing reads)	Brassica spp. with Brassica napus present	Brassica spp. with Brassica napus absent	Vicia spp. with Vicia faba present	Vicia spp. with Vicia faba absent
Predominant (>45%)	37	22	5	0
Secondary (15-45%)	25	8	12	5
Important Minor (1-15%)	28	28	22	10
Minor (<1%)	36	58	33	14
Absent	44	138	101	222

Extended Data Figure 2. The majority of the 47 taxa present changes in their frequency status (from Important minor in 1952 to minor in 2017). How to explain this trend? Is it linked to the fact that DNA metabarcoding is more sensitive to rare species within samples?

When comparing between DNA metabarcoding and melissopalynology the most abundantly found plant taxa were the most consistently detected between the two survey methods, whereas rare taxa were less likely to be detected by both surveys (Hawkins et al., 2015; Richardson et al., 2015). Both survey methods are impacted by the stochasticity in detecting rare plants found at lower levels, but DNA metabarcoding may be more sensitive to taxa at lower levels than melissopalynology, as melissopalynology protocols identify around 300 pollen grains for each sample. To account for this we restrict our comparison to the abundantly found (>15% of sequences/pollen grains) taxa.

Extended Data Fig 4/5/6 : if possible, I would suggest to move them in Supplementary

To also meet the format requirements of Communications Biology the extended data figures have been moved to supplementary.

L.492-507. This section brings important arguments to support the comparative approach between melissopalynology and DNA metabarcoding. This question came to mind all along the reading of the MS, I would thus suggest including some of these information in the method sections if possible. In order to prevent bias in quantitative approaches, the authors analysed the frequency changes of major plants classified as broad categories which sounds to me reasonable but here again, this information deserve to be presented in the MS to understand the figures and the results.

The section discussing DNA metabarcoding compared to melissopalynology has been moved to the methods section at L394.

Then, I am wondering if the authors analysed the pesticide residues in their 2017 honey samples, according to location or habitat for example.

We did not analyse the honey samples for pesticide residues, this would be a very interesting line of further research.

Reviewer #3 (Remarks to the Author):

This paper titled “Shifts in honeybee foraging reveal historical changes in floral resources” analyses historical changes in floral composition based on honey samples. The authors obtained floral resources data using two different methods (DNA metabarcoding and melissopalynology) to detect historical changes.

The authors found a change in the use of major forage plants reflecting agricultural intensification, such as the increase of food crops and invasive species. Overall, the questions and methods used are interesting, and it is clear and well written. I also think that this paper has some merits because it uses historical data to evaluate ecological changes. I do not find, however, the results particularly surprising, but they show the importance of using honey samples to monitor floral resources.

However, I have some major concerns with this study mainly related to the validity of the methods and conclusions drawn from the analysis:

- Comparison between DNA metabarcoding and melissopalynology: Although I see the value of comparing two different data sets obtained from two different methods (and I understand that this is particularly difficult with historical data), I think there are few important limitations and biases that need to be tackled before drawing some relevant conclusions.

The authors acknowledge these limitations in the “Supplementary Discussion” and try to overcome the main problem of doing a quantitative comparison between microscopy and DNA metabarcoding of honey samples by using four broad classes and focusing on the “abundant” (predominant and secondary) classes. This could be a valid approach, however the problem of taxonomic resolution persists. As the authors explain (lines 512-516) some taxa (19) were not identified but the authors do not know if this is because DNA metabarcoding (and may be its associated reference database) was not able to detect (potentially due to other biases (e.g. PCR biases)) or because they are actually not there in those sites. The level of uncertainty is high and hard to control given that there are many confounding factors (e.g. time, location, PCR biases, reference database).

Both DNA metabarcoding and melissopalynology have limitations in their ability to identify all the plants within a honey sample. Both survey methods are impacted by the stochasticity in detecting rare plants found at lower levels. Of the 19 taxa present in 1952 not identified using DNA metabarcoding, nine were likely due to differing levels of taxonomic resolution. For example, the melissopalynology identified several Asteraceae genera which were not found in the DNA, however we did have DNA reads returned that could only be assigned to family level, and potentially the genera identified in 1952 may be contained within these sequences. There were only 10 taxa not found in the DNA which could not be explained by taxonomic resolution. Their absence is not due to the reference library or primers used, as the absent species are ones which can be DNA barcoded successfully. We do not draw any ecological conclusions from their absence in 2017.

To make the decisions in analysis between metabarcoding and melissopalynology clearer in the manuscript, we have moved aspects of the supplementary discussion to the main methods, starting at L394 and added to the results section at L138.

For the comparisons between 1952 and 2017 we focus on species which are reliably detected by both methods, which are the frequently found, abundant species. In the case of taxonomic

resolution for these taxa, there are similar levels of identification between the DNA and the melissopalynology, e.g. *Trifolium pratense* and *Trifolium repens* are able to be distinguished by both the DNA and the microscopy. Both techniques have difficulty in resolving the Maleae group, with hawthorn, apple, cotoneaster grouped together.

I will talk about the location problem specifically in my next point because I have some suggestions to disentangle this effect (see below). Ideally, DNA metabarcoding would have been applied to both 1952 and 2017 samples and/or the study would have gone back to the same sites were 1952 honey samples were taken. Given the constraints of this study, it is hard to make major conclusions without other lines of evidence.

- Geographical/location biases: to understand the effect of space and time, as well as comparing 1952 vs. 2017, the authors did multiple GLMs taking time, latitude and longitude as covariates of the model. My first suggestion is to explicitly put the equations of the GLMs, this always helps to make clear what is the model being analysed besides doing it verbally, it avoids confusions.

Thank you for the advice, for clarification, a single GLM was performed using the “mvabund” package in R which fits generalised linear model to each response variable (species ID) with a common set of predictor variables. We then use resampling to test for significant community level responses to our predictors. We have now clarified this in L381. See formula below.

$$Y = \beta_0 + \beta_1 X_1 + \beta_2 X_2 + \beta_3 X_3 + \log t_x \quad \text{Equation 1}$$

Where β_1 is time measured as month number, β_2 is latitude of the apiary and β_3 is longitude where the honey was collected, t is the number of sequences reads from each site (included in the model as an offset).

Secondly, did the authors check for potential spatial (and temporal) autocorrelation effects? Given that the authors use lat/lons as factors in their models this might violate the assumptions of independence, it might be important to check any spatial autocorrelation effects. For example, besides doing a Moran I test, the authors could plot the residuals versus the lat and lon variable (or even the sampling time of the 2017 honey samples). If there is auto-correlation, a potential solution for this could be to use location as a random effect. I suggest the authors following the regression protocol of Zuur & Ieno 2016 (Methods in Ecol and Evol, 7: 636-645). By doing a GLMM the authors might be able to disentangle some of the confounding factors.

Thank you for your in-depth review of the statistical analysis.

We have produced figures of residuals plotted against latitude, longitude and time and find no correlation (visual inspection of the figures). These figures are now in the supplementary information (Supplementary figure 9A-C).

Following reviewers' suggestions, we have also updated the text for the statistical analysis to make it clear that the model was run a single time with a multivariate response (opposed to multiple glm models performed individually for each taxa). The manyglm modelling

approach is therefore an excellent option for analysing multivariate abundance data produced by metabarcoding, within a single model. Please see changes to statistical analyses section for clarity, L377. Note that it is not possible to add random variables to the manyglm model (like a glmm) however, we are satisfied that there is no autocorrelation in the data and we have modified the manyglm (see response below) to remove the need for scaling.

Minor comments:

- Lines 87-88: Although I see the value of using the countryside survey data, there is a large temporal gap between 1952 and 1978. The data is good to indicate some changes of the flora across the UK, however the authors have to be cautious drawing conclusions about changes of ‘abundance’, especially when comparing with DNA metabarcoding data. I will focus more generally on changes in composition rather than ‘abundances’.

This is a good point, and to make the manuscript clearer ‘abundance’ has been changed to ‘frequency’ for references to the Countryside Survey data as at L152.

- Lines 332-333: I think these transformations are a bit arbitrary; did the authors test other models (e.g. Normal or Gamma distribution) where you don’t need to convert to integers the data?

We tested the data and it best fits a negative binomial distribution; this is also evident from the mean variance relationship in Supplementary Fig. 6 (typical for count data). However, we have now included the evidence that the data best fit a negative binomial distribution and compared with a normal distribution in the supplementary information (Supplementary Fig. 7, Supplementary Fig. 8).

Following reviewers’ suggestions, we have now updated the model to include count data instead of proportional count data. We used proportion of each taxa (relative to the number of sequences in a sample) to control for differences in sequence read numbers from each sample (McMurdie et al 2014). However, we have now included the number of sequences from each sample as a “offset” in the manglm model (Supplementary Fig. 10) which is included in the linear predictor during fitting and removes the need to scale the data. This modification has not changed the conclusions drawn from the model.

- I suggest the authors to compare beta diversity indices between 1952 and 2017 (besides comparing broad class frequencies), and perhaps doing a multivariate analysis (e.g. PERMANOVA) or ordination analyses. These approaches are normally used for metabarcoding data (Laroche et al 2017, PeerJ 5: e3347)

We opted for the model-based approach of mvabund rather than distance-based significance testing methods such as PERMANOVA, in our analysis, as mvabund takes into account the strong mean-variance relationship visualised in supplementary figure 6 (Warton, Wright & Wang, 2012). By using the manyglm, the multivariate abundance data produced by metabarcoding is analysed in one model, rather than as a univariate response.

- Figure 1: could the authors add a map of 1952 honey samples?

In order to provide a comparison between the county level sampling between 1952 and 2017, a heat map of the sampling in both years has been added (Supplementary Figure 5)

- Extended data Figure 5: could the authors add a figure of Pearson residuals versus other covariates (latitude, longitude, time).

We have included three figures showing the relationship between the residuals and latitude (Supplementary Figure 9A), longitude (Supplementary Figure 9B), time (Supplementary Figure 9C).

References

Carvell, C., Roy, D.B., Smart, S.M., Pywell, R.F., Preston, C.D., Goulson, D., 2006. Declines in forage availability for bumblebees at a national scale. *Biological Conservation* 132, 481–489.

Jones L. (2019) Investigating the foraging preferences of the honeybee *L. Apis mellifera* using DNA metabarcoding

Maurizio, A. & Hodges, F. E. D. Pollen Analysis of Honey. *Bee World* 32, 1–5 (1951)),

McMurdie, P. J. & Holmes, S. Waste Not, Want Not: Why Rarefying Microbiome Data Is Inadmissible. *PLoS Comput. Biol.* **10**, (2014).

Preston, C. D., Pearman, D. A., & Dines, T. D. (2002). *New Atlas of the British and Irish Flora: An Atlas of the Vascular Plants of Britain, Ireland, The Isle of Man and the Channel Islands*. Oxford: Oxford University Press.

Warton, D. I., Wright, S. T., & Wang, Y. (2012). Distance-based multivariate analyses confound location and dispersion effects. *Methods in Ecology and Evolution*, 3(1), 89–101. <https://doi.org/10.1111/j.2041-210X.2011.00127.x>

Reviewers' comments:

Reviewer #1 (Remarks to the Author):

Given the strong systematic biases associated with these novel methods, I don't think that comparing historic microscopic data with new molecular data provides for a robust comparison. The works which the authors have referenced to support a more liberal interpretation of the semi-quantitative nature of the data are more complex on a marker-by-marker basis than is acknowledged. The variance explained between molecular and microscopic methods is often quite low (e.g. see Smart et al. 2017). There are also many works which found very poor quantitative performance, which are not referenced for this discussion (<https://doi.org/10.1111/mec.14840>, <https://doi.org/10.1038/s41598-020-57858-2>).

Some of the analytical issues brought forth by review are not addressed. For example, standardizing sampling depth per sample is an important component of ecological experiments. Failing to do so introduces error which could be easily minimized by routine procedures. In another example, objective criteria by which sequence alignments were judged for classification was requested. The authors note that they did not use an objective alignment percent ID cutoff. Instead, they used a bit score threshold. Specifically, 'poor' bit scores were not relied upon. The definition of a 'poor' bit score does not appear to be provided.

Overall, this work represents a attractive hypothesis. I believe it's generally worthy of publication but the underlying data come with high uncertainty.

Reviewer #2 (Remarks to the Author):

I am happy with the corrections the authors made in regards to the reviewer suggestions. In particular, the authors improved the understanding of the manuscript by adding detailed information about the 1952 honey methods, the limitations in melissopalynology and DNA metabarcoding comparison, and they enlarge the honeybee to a broader pollinator community perspective in their discussion. The main caveat remains potential biases in their comparison between 1952 and 2017 datasets due to the lack of precise information (location, season) from the 1952 study and from the differences in the methodology used (melissopalynology versus DNA metabarcoding). The authors can't really change that and these kinds of historical datasets are valuable for the scientific community anyway. Because the authors focused only on the main forage plants, clearly presented the differences in the methods used and discussed the potential limitations in comparing them, and found temporal trends consistent with those found from the CountrySide Survey in plant frequency, I am pretty confident in the results presented in the paper. In addition to this historical temporal aspect, I found the current national view of the plants foraged by honeybees from the 2017 honey analysis with metabarcoding interesting and novel on itself.

L 215-220 Please add the sample size in the methods. This is presented in the results section only.

L 343-352 Why don't you move this paragraph up (L. 326), so that the statistical sections are presented follow.

L. 84-86 and Supplementary Figure 2 . How to explain that you did not find any association between plant composition of the honey and the dominant surrounding habitat class. This is surprising in view

of the differences in plant composition between these habitats. Is it due to a plant 'selection' by honeybees whatever their habitat? Also, a statistical test can be added I think (permanova test with adonis test for example) to support your conclusion.

L.186-191 The authors recommended to increase the amount of flowering clover. This makes sense as they found clover as a major forage plant that have declined and because of the opportunity to increase floral resources at the national scale, given that improved grassland is the dominant habitat in the UK. As suggested by another reviewer, I think the authors need to be cautious in this management recommendation of increases a single genus which if misunderstood, could eventually lead in monospecific grasslands.

Reviewer #3 (Remarks to the Author):

Overall, I think the authors have greatly improved the manuscript by clarifying many aspects of the statistical analyses, data collection and the general interpretation of the results. I find the paper very interesting and insightful showing the advantages of using DNA metabarcoding and honey combined with historical data to monitor terrestrial landscapes and infer historical changes.

Minor comments:

- I suggested the authors to look at the potential problem of spatial autocorrelation and few ways to deal with this problem. The authors have considered some of my suggestions (plotting the residuals) and they did not find a clear effect of latitude and longitude on the response variable(s). However, as I previously suggested in my review, a Moran I test would be a definite way to investigate this problem. I suggest the authors to do the Moran I test and incorporate those results to the Supplementary Materials. The following link provides an example of how to do it in R (doing variograms and using the R package 'spdep'): https://rstudio-pubs-static.s3.amazonaws.com/278910_3ebade4ad6a14f8f9ac6e05eb16b5a21.html

- Lines 100-101: Supplementary Fig 4 shows the frequency of occurrence for 47 plant groups. I suggest to change it to 'taxa' not groups because is confusing.

- Supplementary Fig 2: it is hard to read the labels in the NMDS. I suggest removing it.

- Supplementary Fig 4: in this figure there is a mix of taxonomical levels represented, families (e.g. Poaceae) and genera. It would be clearer for the reader to look at a single taxonomic level, perhaps multiple taxonomic levels can be represented in the figure using other symbols or colours.

- Lines 321-322: What reference collection was used by Deans 1957 to identify the pollen? This information is missing.

Point-by-point response to reviewers

Thank you for the helpful points raised by the reviewers in commenting on our revised manuscript. To address the points raised, we have added further commentary on the biases in comparing the metabarcoding and microscopy data to the manuscript and reanalysed the data with a standardised sequencing depth, demonstrating that it does not change the conclusions we take from the data. We have added further analysis and discussion on the relationship between the plant composition of the honey and the dominant surrounding habitat class. We have also added further statistical testing to investigate potential spatial autocorrelation. The changes made are described in further depth below.

Line numbers refer to the track changes version of the MS (PDF).

Editor's comments

Particularly we ask that you further address the biases in comparing these methods, standardize sampling depth per sample, and provide objective criteria for sequence alignment classification as requested by Reviewer 1.

We have addressed further the limitations in comparing these methods and included the papers suggested by Reviewer 1 at L219.

To address standardising by sampling depth we have presented the results of the analyses if the sequencing data is rarefied as Supplementary Results. We find that the standardised sequencing depth by rarefying re-analysis does not change the conclusions we take from the data. Please see below for a full explanation of how we addressed Reviewer 1's comments in this area.

The criteria used for species identification based on the bitscore percentile has been included at L324, to demonstrate quality checked objectivity in the taxonomic classification.

We also ask that you address the lack of association between plant composition of the honey and the dominant surrounding habitat class as raised by Reviewer 2, and address Reviewer 3's request to further resolve the potential for spatial autocorrelation.

Further statistical analysis and discussion has been added to address the relationship between the plant composition of the honey and the dominant surrounding habitat class at L87, L181, and L479. The relationship was tested by permanova, using the adonis function in R. Please see below for more details.

We have included further analysis to address the potential spatial autocorrelation included at L84 and L392. We compute a Moran's I test as requested by Reviewer 3. Due to the structure of our data we further explore the potential for spatial autocorrelation by testing between the two distances matrices (geographic distance and plant community dissimilarity), examining the

residuals of the spatially explicit model and testing the residuals with Moran's I. Please see below for a full description on our methods.

We therefore invite you to revise and resubmit your manuscript, taking into account the points raised.

Please highlight all changes in the manuscript text file.

Along with this point-by-point response we have included a track-changed version of the original manuscript to enable transparent assessment of our revisions.

Reviewers' comments:

Reviewer #1 (Remarks to the Author):

Given the strong systematic biases associated with these novel methods, I don't think that comparing historic microscopic data with new molecular data provides for a robust comparison. The works which the authors have referenced to support a more liberal interpretation of the semi-quantitative nature of the data are more complex on a marker-by-marker basis than is acknowledged. The variance explained between molecular and microscopic methods is often quite low (e.g. see Smart et al. 2017). There are also many works which found very poor quantitative performance, which are not referenced for this discussion (<https://doi.org/10.1111/mec.14840>, <https://doi.org/10.1038/s41598-020-57858-2>).

Thank you for the references, we have added to the discussion to acknowledge these points and include these papers at L219. We are also mindful of the ongoing debate about whether metabarcoding data should be treated quantitatively or not. In this MS we take a conservative approach to compare the microscopy and molecular datasets. When we compare between the microscopy and the DNA metabarcoding, we follow one of the common recommendations from these papers which is to use the proportion of presence/absence of taxa across the samples. When we compare this frequency of occurrence across samples, we restrict the comparison to only those taxa that are found abundantly within the sample. For both metabarcoding and microscopy, taxa which are abundant within the sample are more reliably detected.

Some of the analytical issues brought forth by review are not addressed. For example, standardizing sampling depth per sample is an important component of ecological experiments. Failing to do so introduces error which could be easily minimized by routine procedures.

The two main approaches to standardising sampling depth per sample are rarefying (i.e. to discard sequences from larger libraries so all samples have the same number of sequences present) or using the proportion of sequences (Deagle et al., 2019). McMurdie and Holmes (2014) criticise rarefying data as a statistical approach when dealing with biological count data since it means losing a considerable amount of biologically valid information. To avoid leaving

out biologically important data we used the proportions of sequences to address the differences in sampling depth.

Following the suggestion of Reviewer 1 we reran the analyses and compared the model outputs after standardizing sampling depth by rarefying the sequencing data. The rarefied data results are included as Supplementary Results at L821.

To explore the impact of rarefying the sequencing data to normalise the libraries, the sequencing sampling depth per sample was standardised by using the *rarefy_even_depth* function in the R package phyloseq. The 15th percentile for library size was chosen ($n = 8012$) and set as the minimum library size, this removed 66 samples from analysis, leaving 375 samples. The statistical analyses presented in the paper were completed again to examine the conclusions.

After rarefying the data, the significant relationship between the frequency of the taxa found in both 2017 and 1952 remained (Kendall's τ correlation coefficient, $\tau = 0.371$, $P = 0.0004$). The differences between those plant taxa found abundantly within the honey samples in both surveys (>15% of DNA sequences or pollen grains within the sample) were then examined. The same patterns of increases and decreases in frequency across the honey samples were seen for the nine taxa as presented. All of the statistically significant changes in frequency remained, with the exception of one plant, *Acer*: ($\chi^2 = 6.853$, d.f. = 1, $P = 0.178$).

The spatial relationship between the presence of *Brassica* and *Vicia* crop species in the landscape with their presence in the honey also remained the same. Apiaries with the related crop species present in a 2 km radius of the hives were more likely to have the crop present in the honey for both *Brassica* spp. and oilseed rape (*Brassica napus*) ($\chi^2 = 45.52$, d.f. = 4, $p < 0.0001$) and *Vicia* spp. with field beans (*Vicia faba*) ($\chi^2 = 48.13$, d.f. = 4, $p < 0.0001$).

The conclusions from the model examining the effects of time and location were unchanged with calendar month (April-October), being a good predictor of plant taxa composition (Fig. 2; $LR_{364, 1} = 427.3$, $P = 0.001$). There were no overall regional differences between England, Scotland and Wales (Supplementary Fig. 1; Latitude $LR_{363, 1} = 229.8$, $P = 0.458$; Longitude $LR_{362, 1} = 324.8$, $P = 0.195$).

Consequently, we see objective value in presenting our full dataset and sample inclusivity which convey the ecological interpretations using all of the evidence available. We now back this up with the analyses of the rarefied data which are described fully in the SI starting at L821 and summarised in the MS at L140.

Deagle, B.E., Thomas, A.C., McInnes, J.C., Clarke, L.J., Vesterinen, E.J., Clare, E.L., Kartzinel, T.R., Eveson, J.P., 2019. Counting with DNA in metabarcoding studies: How should we convert sequence reads to dietary data? *Mol. Ecol.* **28**, 391–406.

McMurdie, P.J., Holmes, S., 2014. Waste Not, Want Not: Why Rarefying Microbiome Data Is Inadmissible. *PLoS Comput. Biol.* **10**.

In another example, objective criteria by which sequence alignments were judged for classification was requested. The authors note that they did not use an objective alignment percent ID cutoff. Instead, they used a bit score threshold. Specifically, 'poor' bit scores were not relied upon. The definition of a 'poor' bit score does not appear to be provided.

Thank you for noting this omission. Sequences with bit scores below the 1st percentile were excluded. This has been added to the methods section at L324.

Overall, this work represents a attractive hypothesis. I believe it's generally worthy of publication but the underlying data come with high uncertainty.

We would like to thank Reviewer 1 for their comments. We have added in some further discussion at L219 to emphasise that the comparison of these data comes with uncertainty.

Reviewer #2 (Remarks to the Author):

I am happy with the corrections the authors made in regards to the reviewer suggestions. In particular, the authors improved the understanding of the manuscript by adding detailed information about the 1952 honey methods, the limitations in melissopalynology and DNA metabarcoding comparison, and they enlarge the honeybee to a broader pollinator community perspective in their discussion.

Thank you for raising these points, their addition improves the manuscript substantially.

The main caveat remains potential biases in their comparison between 1952 and 2017 datasets due to the lack of precise information (location, season) from the 1952 study and from the differences in the methodology used (melissopalynology versus DNA metabarcoding). The authors can't really change that and these kinds of historical datasets are valuable for the scientific community anyway. Because the authors focused only on the main forage plants, clearly presented the differences in the methods used and discussed the potential limitations in comparing them, and found temporal trends consistent with those found from the Countryside Survey in plant frequency, I am pretty confident in the results presented in the paper. In addition to this historical temporal aspect, I found the current national view of the plants foraged by honeybees from the 2017 honey analysis with metabarcoding interesting and novel on itself.

Thank you for your comments. We have added in some further discussion at L219 to emphasise that the comparison of these data comes with uncertainty.

L 215-220 Please add the sample size in the methods. This is presented in the results section only.

This has been added to the manuscript at L242.

L 343-352 Why don't you move this paragraph up (L. 326), so that the statistical sections are presented follow.

Thank you for this note, this has been moved as suggested.

L. 84-86 and Supplementary Figure 2 . How to explain that you did not find any association between plant composition of the honey and the dominant surrounding habitat class. This is surprising in view of the differences in plant composition between these habitats. Is it due to a plant 'selection' by honeybees whatever their habitat? Also, a statistical test can be added I think (permanova test with adonis test for example) to support your conclusion.

This is a very interesting question. We have added a permanova analysis using adonis in R. This showed a significant relationship between the plant composition and the dominant habitat class but explained only a limited amount of the variance in the data (Supplementary Fig. 2; $r^2=0.037$, $P = 0.001$). We have added to the discussion at L181 to reflect this result. The plants we find as most frequently used by honeybees are widely distributed in the UK. Honeybees may be selecting the same frequently found plants across different habitat classes, with time of year being a better predictor for plant choice.

L.186-191 The authors recommended to increase the amount of flowering clover. This makes sense as they found clover as a major forage plant that have declined and because of the opportunity to increase floral resources at the national scale, given that improved grassland is the dominant habitat in the UK. As suggested by another reviewer, I think the authors need to be cautious in this management recommendation of increases a single genus which if misunderstood, could eventually lead in monospecific grasslands.

To remove the potential implication that a monospecific grassland would be the optimal outcome for improving forage, at L199 the recommendation has been changed to advise for increasing the presence and diversity of nectar rich species within improved grassland habitats, *including* flowering clover.

Reviewer #3 (Remarks to the Author):

Overall, I think the authors have greatly improved the manuscript by clarifying many aspects of the statistical analyses, data collection and the general interpretation of the results. I find the paper very interesting and insightful showing the advantages of using DNA metabarcoding and honey combined with historical data to monitor terrestrial landscapes and infer historical changes.

Thank you ever so much for your comments which have greatly improved the paper.

Minor comments:

I suggested the authors to look at the potential problem of spatial autocorrelation and few ways to deal with this problem. The authors have considered some of my suggestions (plotting the residuals) and they did not find a clear effect of latitude and longitude on the response variable(s).

However, as I previously suggested in my review, a Moran I test would be a definite way to investigate this problem. I suggest the authors to do the Moran I test and incorporate those results to the Supplementary Materials. The following link provides an example of how to do it in R (doing variograms and using the R package 'spdep'): https://rstudio-pubs-static.s3.amazonaws.com/278910_3ebade4ad6a14f8f9ac6e05eb16b5a21.html

Thank you for this suggestion. To investigate the relationship between latitude, longitude and date with the plant composition of the honey, we used a multivariate model with the manyglm function. Within this model we are looking at the relationship of the variables with each taxa's abundance within the honey, rather than a single metric (e.g. species richness). Therefore to test for spatial autocorrelation between the geographic distances and the community composition differences we think it is appropriate to use a Mantel test and compare between these two distance matrices. To test for spatial autocorrelation between the plant community composition and hive location a Mantel test with 9,999 permutations was used. The Bray-Curtis dissimilarity index was used to estimate β -diversity. The community dissimilarity was not found to change with distance between sampling sites ($r = 0.006$ $P = 0.387$). We have reported this at L86 and L392.

We have used a spatially explicit model and examined the relationship between latitude, longitude and date with the plant composition of the honey and found that very little variation was explained from latitude and longitude (Supplementary Fig. 1; Latitude $LR_{427,1} = 272.2$, $P = 0.086$; Longitude $LR_{426,1} = 352.3$, $P = 0.092$).

To test for SAC within the residuals we took the average residual value for each location and computed Moran's I (Moran I statistic standard deviate = 1.638, $P = 0.05071$), reported at L397.

Model Residuals:

Lines 100-101: Supplementary Fig 4 shows the frequency of occurrence for 47 plant groups. I suggest to change it to 'taxa' not groups because is confusing.

The wording 'groups' has been changed to taxa to prevent confusion as suggested at L106.

Supplementary Fig 2: it is hard to read the labels in the NMDS. I suggest removing it.

To make the figure clearer, the species labels have been removed. The figure has been left in to support the additional discussion suggested by Reviewer 2.

Supplementary Figure 2: Non-metric multidimensional scaling (NMDS) ordination of the 2017 honey samples collected in July and August. Colour indicates the dominant surrounding habitat measured within a 2 km radius of the hive location.

Supplementary Fig 4: in this figure there is a mix of taxonomical levels represented, families (e.g. Poaceae) and genera. It would be clearer for the reader to look at a single taxonomic level, perhaps multiple taxonomic levels can be represented in the figure using other symbols or colours.

The points and taxa labels in Supplementary Figure 4 have been coloured by taxonomic rank to address this as suggested.

Supplementary Figure 4: Comparing the total proportion of samples found in 1952 and 2017 for the plant taxa found in both surveys. There is a significant positive correlation (Kendall's τ correlation coefficient $\tau = 0.389$, $P < 0.001$). Taxa which appear in over 10% of samples for either the 1952 or 2017 survey are labelled.

Lines 321-322: What reference collection was used by Deans 1957 to identify the pollen? This information is missing.

Further information on the reference libraries reported by Deans has been added to the manuscript at L353.

Response to Reviewers

Dear Dr. de Vere,

I apologize for the delay in returning a decision on your manuscript entitled "Shifts in honeybee foraging reveal historical changes in floral resources" for Communications Biology, and I wanted to provide you with an update. Your manuscript has now been seen again by 2 referees, one of which was an original reviewer for the previous versions of the manuscript, Reviewer 3, and an additional Reviewer 4. I have amended both of these reports below.

While Reviewer 3 generally finds the manuscript improved from the previous version, this reviewer still has concerns regarding the analyses to address spatial autocorrelation and requests the use of a distance-based Moran's eigenvector map analysis, which we do consider necessary to fully address these concerns. However, in order to avoid an additional round of review, we ask that you perform this analysis, in addition to addressing the other concerns of both reviewers, before we make an official decision on the manuscript.

Thank you to the reviewers for their comments. We have amended the manuscript as suggested, performing the distance-based Moran's eigenvector map analysis, in addition to making the other suggested comments and changes. The changes we have made are detailed below. The line references refer to the PDF version of the MS.

Reviewer Comments

Reviewer 3 Comments to the Author:

The authors have addressed my comments about spatial autocorrelation in different ways. I appreciate the efforts by the authors dealing with this issue. The manuscript has been greatly improved. I do have, however, some minor concerns/comments related to what they did for this specific point:

- The authors performed a Mantel test to test for spatial autocorrelation between plant community composition and hive location using a Bray-Curtis dissimilarity matrix to estimate beta-diversity. They did not find any change of beta-diversity with geographic distance (as explained in rebuttal letter). Regarding this analysis it was a bit surprising not finding any effect of geographic distance with species turnover if you consider other findings of the paper (for example, Suppl Fig 1). The use of Mantel tests for this issue can be problematic sometimes (see Legendre, Fortin and Borcard (2015) *Methods in Ecol and Evol*) particularly when violating assumptions of the Mantel test such as linearity and homoscedasticity. As suggested by Legendre et al (2015), I highly recommend using distance-based Moran's eigenvector map (MEM) analysis to have a more robust answer.

The results of the Mantel test have been removed as suggested (at Line 95 and Line 408) and replaced with a spatial analysis using distance-based Moran's eigenvector maps. This has been added to the MS at Line 90, 204 and 403. The table of Moran's I's results against individual taxa have been fully reported in the Supplementary Data.

Methods: "We completed a spatial eigenfunction analysis using distance-based Moran's eigenvectors. Moran's Eigenvector Maps were computed using the 'mem' function from the *adespatial* package. Moran's I was computed for each taxa using the 'moran.randtest', with Bonferroni correction for multiple testing. The direction of autocorrelation (positive and negative) was tested using the 'moranNP.randtest' function, using the *adespatial* package in R. "

Results: “While latitude and longitude were not significant predictors when assessing the overall honey composition, at the individual taxa level there was some evidence of spatial autocorrelation in 22 of the 157 taxa identified (using Moran’s I; Supplementary Data). However, after Bonferroni’s correction for multiple testing none of the 22 taxa remained significant.”

Discussion: “While the overall plant composition of the honey was found to be unrelated to location of the hive, further work could investigate the potential geographic patterns present in the spatially restricted plant species found at lower levels within the honey.”

We would like to note that our original statistical tests within this MS already include a spatial component. This is found to be non-significant within our model. We have now performed a wide variety of spatial analyses – none of which find a significant spatial relationship with overall honeybee plant use. This result makes sense given what we know about honeybee foraging – both from this MS, our other research in this area (listed below) and that of other authors.

<https://journals.plos.org/plosone/article?id=10.1371/journal.pone.0134735>

<https://www.nature.com/articles/srep42838>

The work within this area is beginning to reveal a picture of honeybee foraging. Honeybees visit many types of plants but actually most of their diet appears to be composed of a relatively small number of very common plant species. These plants have a wide distribution and are abundant in a wide range of habitat types. As such it makes sense that there is no overall geographic pattern to honeybee foraging within the UK.

Other minor comments:

- Suppl Fig 1: what is the reason to show these results based on regions/political boundaries instead of showing it by type of habitat or bioregion? Also, the number of sampling sites per region/country is highly unbalanced, what is this figure really telling us? I think this figure would be more insightful if it shows us whether pollen DNA metabarcoding results actually reflect differences of plant communities/habitat (or other external drivers) between for example, urban/agricultural/natural habitats or type of vegetation (e.g. grassland, heaths,

woodland) or habitat class. This will partly demonstrate the robustness of your overall results. I suggest redoing this figure considering my suggestion.

We investigate the relationship between the dominant surrounding habitat class and the plant composition of the honey in Suppl Fig 2.

Suppl Fig 1 is there to provide a broad overview of the sites sampled within the MS. The distribution of samples is unbalanced between the regions since samples were provided by contacting beekeepers and asking them to send us samples.

The position of beehives is not a natural feature, but instead is chosen by the beekeeper and these beekeepers are organised into local, regional and national associations. The engagement of the different associations will have influenced the number of honey samples sent to us, so we feel it is relevant to show the distribution of samples using regional/political boundaries.

- As an additional comment to my previous comment, based on the PERMANOVA results and discussion in line 181, it looks as the authors found that honeybees prefer to forage widely distributed plants, this somehow questions the usefulness of honeybees as ideal models to assess landscape changes (lines 35-37). Is it perhaps that several of the key plant species from each habitat class are discarded during the analysis (low number of sequences) while these widely distributed species are preferentially amplified?

During analysis of the sequencing data sequences found at low levels are not removed during the metabarcoding analysis. The PERMANOVA analysis looks at the relationship between the total plant composition of the honey and the dominant habitat type surrounding the hive, to look for any general association. This includes both the plant species found in the majority of the honey and those found at a low level.

In our discussion we have concentrated on the plants most frequently and abundantly found to allow comparison with the 1952 samples. The changes in availability of these widely distributed forage plants will have landscape level impacts which is what we refer to at Line 37 when we say “assess landscape changes in forage availability”.

- Lines 86-89: R-squared of 3% is too low to be considered a 'positive association', despite being significant. I suggest removing the words 'positive association' in this result.

The words 'positive association' have been removed, and changed to "the relationship between the plant composition of the honey and the dominant surrounding habitat class was significant, however habitat class explained only 3% of the total variation" at line 96.

Reference:

Legendre P, Fortin M-J, Borcard D (2015) Should the Mantel test be used in spatial analysis?. *Methods in Ecology and Evolution*, 6, 1239– 1247.

Reviewer 4 Comments to the Author:

This manuscript compares the source plants of honey samples collected in 1952 and in 2017 throughout UK and discusses changes in honey bee foraging resources in the context of landscape changes over this time span. It is an interesting approach to examine the difference in honey collected 65 years apart and link the data with changes in landscape and plant community reported by the Countryside Survey. The amount of data presented is impressive and extremely valuable for both honey bee researches and pollinator conservation.

As the previous reviewers have pointed out, the biggest weakness of this work is the discrepancy between two very different methods: traditional microscopy for 1952 samples vs. DNA metabarcoding for the 2017 samples. I wish the authors had analyzed a subset of the honey samples with both methods as a measure of calibration to allow for better comparisons. The authors used the presence/absence of taxa in samples instead, which I think is reasonable given the large number of samples.

This discrepancy between methods doesn't necessarily prohibit publication of this work but I think the authors need to more transparent about it early in the manuscript. I'd suggest pointing out that DNA metabarcoding has higher resolution than microscopic melissopalynology (and therefore a more conservative approach was used to compare data) where both methods are mentioned (Line 48), before results are presented.

Thank you for this suggestion, a line has been added at line 48 to highlight that we are comparing between two differing techniques.

Another issue is that the authors seem to overinterpret data at times. For example, the discussion about the increase of *Impatiens glandulifera* when it's only detected in a small number of plots. Also it's probably better to err on the side of caution when discussing species-level results e.g. *Rubus fruticosus* and *Vicia faba*, unless they can be reliably detected to the species level. *Rubus* and *Vicia* are both listed as spp. in the data file, not the species.

The reference to the *Impatiens* countryside survey plots has been removed at line 146. Please see below for discussion about *Rubus* vs *Rubus fruticosus*.

I was confused by the term "frequency" in the comparison of 1952 vs. 2017 samples when I first read the manuscript. It made a lot more sense after I read the response reviewers and realized it was the presence/absence of plant taxa across all samples. Please clarify how "frequency" was calculated (Line 59).

This line has been amended as suggested below to: "the total frequency of occurrence for each plant taxon was calculated as the presence of the taxon across all 2017 honey samples."

Another term needing clarification early on is "major taxa". In Results (Lines 62-67) The plant taxa in a sample was defined as "predominant", "secondary", etc. according to their relative abundance in the sample. The terms "major taxa" and "major forage" are also used throughout the manuscript to describe taxa present in high abundance but was not defined until Line 364. Please specify that "major taxa" means predominant and secondary taxa in Results. I think "major taxa" should be used consistently and not interchangeably with "major forage". This study examined the abundance of sequences/pollen grains but that information doesn't really translate to the quantity of honey coming from given taxa due to the biology of the plants and pollen filtration in the bee's digestive tract. See Bryant & Jones 2001 (ref. 50).

Thank you for this suggestion. A definition for major taxa has been added to the results section and where “major forage” was used, this has been replaced with “major taxa”.

More specific comments below:

Lines 59-60: “The total frequency of occurrence for each plant taxon was calculated [as the presence of the taxon] across [all 2017] honey samples.”

Thank you for improving the clarity, this line has been amended as suggested (Line 64-65).

Line 68: Figure 2 only shows *Rubus* spp. But here you say *R. fruticosus* was the most frequently found and abundant plant in honey samples. Were you able to detect *Rubus* at the species level? What’s the abundance of *R. fruticosus* within the *Rubus* spp. detected?

DNA metabarcoding can only identify *Rubus* to genus level. Combined with this data we have incorporated the botanical knowledge of the UK. Within the UK, *Rubus fruticosus* agg. is a widespread, common species. The most common alternative native species present are *R. chamaerorus* (cloudberry), *R. caesius* (dewberry), and *R. idaeus* (raspberry), which are much more restricted in distribution, either to peaty moors, bogs on mountains, or coastal dune slacks. Raspberry can be more common as a cultivated species.

This is illustrated within the Countryside Survey data where in 2007 *R. fruticosus* was found in 341 plots, *R. chamaerorus* in 10, *R. caesius* in 5, and *R. idaeus* in 16. While we do not want to overstate the ability of the DNA metabarcoding’s ability to identify to species level, we think it appropriate to assume that the distribution and availability of *R. fruticosus* is contributing the most to the *Rubus* DNA detected within the honey.

To clarify this within the manuscript we have noted that the metabarcoding results identify to *Rubus* spp. and that we have compared this with the most widely distributed and common *Rubus* species, *Rubus fruticosus* starting at line 134.

Lines 100-101: Did the authors compare results using both microscopy and DNA metabarcoding on the same honey samples? The supplemental discussion only talks about taxa found in the 1952 samples but not in 2017 samples (and vice versa). There is no way to tell if the difference was due to the presence/absence of pollen or the ability to correctly

identify these taxa with each method. Unless you have analyzed the same set of samples using both methods, I'd suggest moving this statement about DNA metabarcoding leveraging greater taxonomic resolution than melissopalynology up to Line 48 as I mentioned above.

As suggested above, the sentence indicating that DNA metabarcoding leverages greater taxonomic resolution compared to melissopalynology has been added to Line 49.

"DNA metabarcoding leverages greater taxonomic resolution of the plant taxa present in the honey when compared to microscopic identification and so a conservative approach was taken to compare between the data."

Line 97-98: Please include citations for the 1952 study by Deans in the first sentence.

Thank you for noting this omission, the citations have been added.

Line 136: "Impatiens glandulifera increased by 100% ..." In Figure 4, the number of plots where *I. glandulifera* was present only change from 2 plots in 1978 to 4 plots in 2007, out of the 1577 fixed plots. Emphasizing the 100% increase is a bit excessive for describing the 2-plot difference.

The sentence over emphasising the 100% increase for this limited number of plots has been removed.

Lines 168 – 171: I don't see how you came to the conclusion that honey bees will use *Vicia faba* as the major forage where the crop is grown. Could DNA metabarcoding detect *V. faba* to the species? The presence of *Vicia* spp. decreased from 32% to 23% of honey samples, which reflects the decline in *Vicia* spp. in the countryside survey. *Vicia* was the major taxa in slightly more honey samples (increase from 2% to 5%), but the percentage is still quite small. Figure 3 also shows over 50% of samples containing no *Vicia* pollen in the presence of field bean crop. I find this part of the discussion confusing but it's a minor issue.

This part of the discussion starting at Line 179 has been reworded for clarity and to indicate that the increase in *Vicia* as a major forage may be potentially explained by the increase in availability of field beans as a crop.

Methods for 1952 Honey Sampling (Lines 332 – 348):

Did the authors analyze the 1952 samples or is this paragraph describing Deans' methods? The way this information is presented sounds like the authors did the analysis (or is Deans a co-author?). If not, you need to clarify that or perhaps include this part as a supplemental material?

To provide clear ease of access to Dean's methods for the 1952 honey survey and so allow readers to be able to compare between the two techniques we have reported, we have described Dean's methods as fully as possible within the MS. To clarify that this work was undertaken separately, we have added a sentence stating: "The methods reported for the research conducted in 1952 are described here fully for comparison" at Line 358.

Figure 4: Update "Countryside Survey Plot Frequency" to reflect the data presented, which is the number of plots where given plant taxa were detected. Frequency would be the % of plots where they were detected.

Unless *R. fruticosus* was detected at the species level, change it to *Rubus* spp.

The title "Countryside Survey Plot Frequency" has been changed to "Countryside Survey Plot Count" and *Rubus fruticosus* to *Rubus* spp. as shown below for Figure 4.